# CO<sub>2</sub> variability and seasonal cycle in the UTLS: Insights from EMAC model and AirCore observational data

Johannes Degen<sup>1</sup>, Bianca C. Baier<sup>2</sup>, Patrick Jöckel<sup>3</sup>, J. Moritz Menken<sup>3</sup>, Tanja J. Schuck<sup>1</sup>, Colm Sweeney<sup>2</sup> and Andreas Engel<sup>1</sup>

- Institute for Atmospheric and Environmental Science, Goethe University Frankfurt, Frankfurt/Main, Germany Global Monitoring Laboratory, National Oceanic and Atmospheric Administration, Boulder, CO, USA Deutsches Zentrum für Luft- und Raumfahrt, Institut für Physik der Atmosphäre, Oberpfaffenhofen, Germany
- 10 Correspondence to: Johannes Degen (degen@iau.uni-frankfurt.de)

## Abstract.

The complex distribution of CO<sub>2</sub> in the upper troposphere and lower stratosphere (UTLS) results from the interplay of different processes and mechanisms. However, in such difficult-to-access regions of the atmosphere our understanding of the CO<sub>2</sub> variability remains limited. Using vertical trace gas profiles derived from measurements with the balloon-based AirCore technique for validation, we investigate the UTLS and stratospheric CO<sub>2</sub> distribution simulated with the ECHAM/MESSy Atmospheric Chemistry (EMAC) global chemistry-climate model. By simulating an artificial, deseasonalised CO<sub>2</sub> tracer, we disentangle the CO<sub>2</sub> seasonal signal from long-term trend and transport contribution. This approach allows us to study the CO<sub>2</sub> seasonal cycle in a unique way in remote areas and on a global scale. Our results show that the tropospheric CO<sub>2</sub> seasonal cycle propagates upwards into the lowermost stratosphere and is most modulated in the extra-tropics between 300-100 hPa, characterised by a 50 % amplitude dampening and a 4-month phase shift in the northern hemisphere mid-latitudes. During this propagation the seasonal cycle shape is also tilted, which is associated with the transport barrier related to the strength of the subtropical jet. In the stratosphere, we identified both, a vertical and a horizontal 'tape recorder' of the CO<sub>2</sub> seasonal cycle. Originating in the tropical tropopause region this imprint is linked to the upwelling and the shallow branch of the Brewer-Dobson-circulation. As the CO<sub>2</sub> seasonal signal carries information about transport processes on different timescales, the newly introduced tracer is a very useful diagnostic tool and would also be a suitable metric for model intercomparisons.

## 1 Introduction

Understanding the atmospheric distribution of carbon dioxide (CO<sub>2</sub>) is not only relevant in the context of climate change, but CO<sub>2</sub> is also a useful tracer for atmospheric transport and dynamics. Since there are potential changes to atmospheric circulation induced by the increase of greenhouse gases (GHG), these aspects are closely interlinked. The upper troposphere and lower stratosphere (UTLS) is a highly climate-sensitive region of the atmosphere. As there is a relative minimum of temperature in this region, changes in the structure and chemical composition of the UTLS result in particularly large changes in radiative forcing (Solomon et al., 2010; Riese et al., 2012). While the transition zone of the UTLS is a mixture of air masses from different origins (Gettelman et al., 2011), it is characterised by complex patterns and sharp gradients of many trace gases, controlled by transport, chemical and microphysical processes on varying spatial and temporal scales. Changes in atmospheric temperature potentially feed back onto bidirectional exchange processes in the UTLS. In addition, the UTLS is influenced by the global-scale stratospheric overturning circulation, known as the Brewer-Dobson circulation (BDC), which is itself a major component of the climate system. It is expected that the strength of the BDC will be affected by climate change (Butchart, 2014), because shifts in temperature and modification of wave propagation will affect the tropical upwelling mass flux (Butchart et al., 2010; Abalos et al., 2021). While models consistently predict an acceleration of the BDC, this remains still unclear from observations (Engel et al., 2017).

Because of its long lifetime, CO<sub>2</sub> can be used to study mechanisms and the interplay of processes controlling its distribution in the UTLS and above. It is basically chemically inert in the free troposphere and has only a minor stratospheric source (<1 ppm) from methane oxidation (Boucher et al., 2009).

As part of the complex global biogeochemical carbon cycle, CO<sub>2</sub> is exchanged between the reservoirs of the atmosphere, ocean, biosphere, and lithosphere. The timescales of these circulation processes range from sub-daily (e.g. photosynthesis/respiration, air-sea gas exchange) to millennial (e.g. deep ocean pool) (Friedlingstein et al., 2024), while exchange with the geological reservoir occurs over even longer periods of time (Archer et al., 2009). The atmospheric CO<sub>2</sub> variability is not only a function of natural sources and sinks, it is also determined by human perturbations. Since the beginning of the Industrial Era (~1750), CO<sub>2</sub> is globally increasing, due to land-use change activities (especially deforestation) and fossil fuel burning. The higher total CO<sub>2</sub> content, combined with effects of the resulting global warming, in turn modifies, for example, the seasonal cycle of CO<sub>2</sub> ("amplification"; Forkel et al., 2016), which is mainly controlled by carbon uptake and release processes of the terrestrial biosphere (Keeling et al., 1996).

Because CO<sub>2</sub> observational data become more and more sparse with increasing distance from the Earth's surface, there are few investigations of the seasonal variation of CO<sub>2</sub> above the boundary layer, and our knowledge of the vertical distribution of CO<sub>2</sub> is still insufficient. Despite their great potential in terms of data coverage and recent improvements in retrieval techniques, satellite-based observations of CO<sub>2</sub> are (at least for now) of limited use in this context. Most studies of CO<sub>2</sub> from space borne sensors focus on total column-averaged dry air mole fractions (XCO<sub>2</sub>) (e.g. Antezana Lopez et al., 2025), due to the characteristics of the nadir sounding instruments that equip the major CO<sub>2</sub> monitoring satellites such as NASA's Orbiting

Carbon Observatory (OCO) and Greenhouse Gases Observing Satellite (GOSAT). Although limb viewing sounders, like the Atmospheric Chemistry Experiment Fourier Transform Spectrometer (ACE-FTS), already offer a much better vertical resolution for CO<sub>2</sub> of about 2.5 km in the UTLS (Foucher et al., 2011), this is in a completely different order of magnitude compared to in situ observations, which also applies to the significantly poorer accuracy of roughly 2 ppm.

Localised but high resolution aircraft measurements including CO<sub>2</sub> in the free troposphere and above are available from campaigns like SPURT (Engel et al., 2006) or DCOTSS (Gordon et al., 2024) and long-term projects such as CONTRAIL (Machida et al., 2008) and IAGOS-CARIBIC (Schuck et al., 2009). These data have been used to analyse the temporal and spatial variability of CO<sub>2</sub> in the UTLS, focusing either on the seasonal CO<sub>2</sub> cycle as a function of latitude (Nakazawa et al., 1991), on the seasonal CO<sub>2</sub> variability at different distances from the tropopause (Gurk et al., 2008; Sawa et al., 2008) or on inter-annual variations in CO<sub>2</sub> distribution patterns (Matsueda et al., 2002). The gradients in CO<sub>2</sub> distribution and changes in the seasonal cycle of CO<sub>2</sub> are helpful tools for investigating transport in the UTLS (Boering et al., 1996; Strahan et al., 1998). They have been used to diagnose stratosphere-troposphere exchange (Strahan et al., 1998; Gurk et al., 2008; Assonov et al., 2013), lowermost stratosphere (LMS) mixing (Hoor et al., 2004; Pan et al., 2006) or the connection of the tropics and extra tropics in the UTLS region including the role of the subtropical jet (STJ), and even to derive transport timescales (Bönisch et al., 2009; Miyazaki et al., 2009). Moreover, several studies brought aircraft data together with model simulations to either better understand the formation mechanisms of CO<sub>2</sub> gradients (Strahan et al., 1998; Miyazaki et al., 2009; Diallo et al., 2017) or to evaluate the model representation of the underlying processes controlling the distribution of CO<sub>2</sub> (Bönisch et al., 2008; Bisht et al., 2021).

However, airborne data on the chemical composition is comparatively expensive and limited in terms of vertical coverage. Using passenger aircraft for making atmospheric observations as with IAGOS-CARIBIC or CONTRAIL restricts the sampling to regular/typical flight altitudes (9-12 km) and, with a few exceptions (NASA WB-57, ER-2), most of the state-of-the-art research aircraft (e.g. HALO, HIAPER) rarely reach altitudes higher than 14 km. High-altitude balloons are therefore the best available way to get in situ trace gas observations above approx. 16 km. In contrast to occasional high-altitude large-balloon campaigns (Nakazawa et al., 1995; Moore et al., 2003; Engel et al., 2009; Schuck et al., 2025) the AirCore technique offers a relatively simple and cost-effective, but equally high-reaching and high-resolution sampling option.

The idea to obtain continuous vertical trace gas profiles from a balloon-borne air sampling system called AirCore was originally proposed by Tans (2009). Ambient air is passively collected in a long and narrow stainless steel tube, that is open at one and closed at the other end, taking advantage of the increasing atmospheric pressure during descent from high altitude to the ground. To preserve vertical structures and to archive the best possible resolution, optimizing AirCore geometry and measuring the sample as soon as possible after recovery minimizes the averaging effects of mixing and molecular diffusion of the sample (Karion et al., 2010). Most commonly the AirCore sample is analysed with a continuous-flow gas analyser for CO<sub>2</sub>, CH<sub>4</sub> and CO.

In recent years, developments of the technique such as adapted designs to provide optimum resolution in the stratosphere (Engel et al., 2017) or configurations with extra high-resolution (Membrive et al., 2017; Laube et al., 2025) took place. For

example, the work of Wagenhäuser et al. (2021) on testing altitude attribution processes with a spiking method and the discussion of the treatment of AirCore filling dynamics by Tans (2022) contributed to advanced concepts and innovations in the way the final trace gas profile is derived from measurements. Validation and inter comparison campaigns continue to ensure that the relatively new AirCore technology has become firmly established. Since standard lightweight AirCores (usually < 3.5 kg) can be easily launched aboard a regular weather balloon, neither safety concerns nor high costs prevent routine measurements, which have been carried out at least for major GHG for several years. Although the overall coverage is still quite sparse and each flight represents only a 'snapshot' of atmospheric conditions at a given location, a growing database of AirCore vertical profiles for CO<sub>2</sub>, CH<sub>4</sub> and CO is now available (e.g. Baier et al., 2021).

Such AirCore profiles provide unique and valuable observational data. They are extremely useful for the evaluation of remote sensing methods such as ground-based Fourier Transform Infrared (FTIR) spectrometers (Sha et al., 2020; Zhou et al., 2023) or satellite retrievals (Tu et al., 2020; Martínez-Alonso et al., 2022). In addition, information from AirCore can be used to monitor the stratospheric circulation and its potential changes (Moore et al., 2014; Engel et al., 2017; Ray et al., 2024). AirCore data are also well suited to play an important role in model evaluation (Diallo et al., 2017).

The accurate simulation of CO<sub>2</sub> in models is particularly challenging because of the superposition of short-term variations (seasonal variability) and long-term changes. However, provided they have been evaluated, results from sophisticated Earth system models can strongly contribute to our understanding of trace gas patterns and underlying processes, especially in remote and difficult-to-access parts of the atmosphere like the UTLS and the stratosphere. With their wide coverage in space and time, they complement the limited but high resolution (in situ) observations available. In addition, models offer the possibility of simplifications or auxiliary tools for investigation, such as idealized and artificial tracers.

EMAC (ECHAM/MESSy Atmospheric Chemistry) is a state-of-the-art modular global climate and chemistry simulation system with submodels for tropospheric and middle atmosphere processes, their interactions with the ocean and land surface, as well as anthropogenic forcing (Jöckel et al., 2005, 2016). EMAC was included in multi-model intercomparisons such as the Chemistry Climate Model Initiative (CCMI, Morgenstern et al., 2017), and the results show that relevant characteristics (e.g. stratospheric circulation) compare well with other climate models (Dietmüller et al., 2018).

In this paper, we use a compilation of vertical trace gas profiles derived from measurements with the balloon-based AirCore technique with EMAC model results to investigate the distribution and variability of CO<sub>2</sub> in and around the UTLS. Based on vertical profile comparisons, we evaluate whether EMAC can adequately map the distribution of CO<sub>2</sub> in the atmosphere and elaborate uncertainty factors for the simulation. Going beyond the discussion of the characteristics of the CO<sub>2</sub> distribution, we focus on patterns in the CO<sub>2</sub> seasonal cycle. A novel approach using a combination of various EMAC CO<sub>2</sub> products including an artificial deseasonalised CO<sub>2</sub> tracer allows us to isolate the CO<sub>2</sub> seasonal signal from transport contribution and long-term trend, so that we can address the question of how the seasonality of CO<sub>2</sub> in the free troposphere propagates into the LMS. Based on a detailed view on the modulation of this CO<sub>2</sub> seasonal signal in the extratropical UTLS and followed by a more global perspective of its distribution, we formulate implications for large-scale transport processes and time scales.

## 2 Data and methods





# 2.1 AirCore technique and available observational data

In principle, an AirCore is a long thin-walled stainless steel tube which is attached to a balloon to collect an air sample (Karion et al., 2010). During ascent, the tube, which is previously filled with a well-characterised fill gas (FG) evacuates, because it is opened at one, but closed on the other end. Upon descent, air is passively sampled and continuously pushed into the AirCore due to the now increasing ambient pressure. If the AirCore is recovered quickly after landing, so that the air inside is only partially mixed and the information of the vertical distribution is retained, the sample can be analysed for multiple trace gas species with a continuous flow gas analyser by pushing it through the coil with a well-characterised push gas (PG). Meteorological and position data recorded simultaneously during the flight are assigned to the trace gas measurements to derive a vertical profile.

An increasing number of AirCore flights is carried out by organizations worldwide, due to the possibility to quite easily launch these lightweight samplers at relatively low cost. Besides technical improvement and intercomparison campaigns aiming for ensuring data quality, there are efforts for the standardization and approaches to establish continuous measurement series.

For this study we build up a new database of AirCore vertical trace gas profiles. This includes the latest version of the openaccess National Oceanic and Atmospheric Administration (NOAA) AirCore atmospheric sampling system profiles dataset (Version 20240909; Baier et al., 2021), a variety of AirCore data from previous campaigns in which Goethe-University Frankfurt (GUF) was involved (Engel et al., 2017; Schuck et al., 2025), and new profiles from the ongoing regular bi-monthly GUF AirCore flight time series from Frankfurt, Germany, starting in September 2022. In addition to the revision of the profiles by each respective research group (e.g. to identify and exclude contaminated parts of the profiles due to leakages, problems with closing valve, etc.), we applied standardized filters to ensure plausibility and uniform treatment of the data.

in a radius of 30-100 km around the launch site (depending on wind conditions during flight and descent speed). AirCore samples were analysed with a Picarro G2401 cavity ring down spectrometer for CO<sub>2</sub>, CH<sub>4</sub> and CO. This instrument is able to measure both, CO<sub>2</sub> and CH<sub>4</sub> with a very high precision (0.01 % for CO<sub>2</sub>, 0.05 % for CH<sub>4</sub>). Altitude retrieval was either performed based on a pressure-equilibrium approach (GUF flights, Engel et al., 2017; Membrive et al., 2017; Wagenhäuser et al., 2021) or according to a fluid-dynamics approach (NOAA flights, Tans, 2022). The trace gas measurements were calibrated using the analyser specific offset and slope values as well as daily offset correction from measurements against a standard calibration tank and the results are reported as dry mole fractions on WMO scales (CO<sub>2</sub>: X2019, CH<sub>4</sub>: X2004A, CO: X2014A).

In most cases the AirCores collected a sample for about 30-45 minutes, reaching up to approximately 30 km with a trajectory

Moreover, a so called 'FG correction' was performed based on the approach described by Engel et al. (2017). This calculates the fraction of FG remaining in the AirCore at the topmost portion of the profile and corrects the first sampled stratospheric air that has mixed with FG based on the fact that the FG is spiked with a tracer (here CO). The fraction of FG is defined by the expected stratospheric background mole fraction of CO, and is either determined by the lowest amount of CO that is measured in the profile (NOAA) or the mean CO value between 80 hPa and 100 hPa (GUF).

Figure 1 shows an example for the vertical trace gas profiles derived from such a single AirCore flight (Flight 2024-03-26 with GUF005 AirCore from Frankfurt, Germany) next to the supporting information on ambient temperature and humidity. Uncertainties are shown as shaded areas. Note that they are strongly dominated by the uncertainties from altitude assignment and that the uncertainties from analytics are not really visible as they are usually smaller than symbol size. The grey dots represent data affected by mixing with either FG (stratospheric end) or PG (tropospheric end). In the upper part of the profile the effect of the 'FG correction' becomes visible. As soon as the PG or FG fractions are too large, the data is excluded from the final analysis.

**Figure 1.** Vertical profiles of CO<sub>2</sub> (a) and CH<sub>4</sub> (b) derived from AirCore flight 2024-03-26 from Frankfurt, Germany. Uncertainties are mainly from altitude mapping and are shown as shaded areas. Grey dots represent measurements affected by mixing with FG or PG. The meteorological information (c) comes from a simultaneously flown radiosonde on the same balloon string and the image in (d) shows the balloon launch.



In total, the combined dataset contains 216 vertical trace gas profiles. They were derived with slightly different setups with more than 20 individual AirCores. Nevertheless, the majority of the flights was either performed with a GUF AirCore (see Engel et al., 2017; 13 % of the profiles), an AirCore according to the standard NOAA design (67 %) or NOAA StratoCore design (18 %) (Karion et al., 2010; Li et al., 2023). Technical details of the different AirCore setups are summarised in supplementary Table S1. Typical values for the vertical resolution range from around 1000 m in the lower stratosphere to less than 300 m in the UTLS and the troposphere (Karion et al., 2010; Engel et al., 2017). Further information on the role of AirCore geometry and discussion on other factors influencing AirCore resolution is provided by Karion et al. (2010) and Tans (2022).

Moreover, details on the exact methodology and specifications for each individual AirCore flight is given as part of the NOAA and GUF data sets that are available from Baier et al. (2021) and Degen et al. (2025).

The spatial and temporal coverage of all those AirCore observational data is visualised in Fig.2. Panel (a) shows a map of all flight locations. Most of the launches took place in the west central part of the U.S. (mainly Colorado). In addition, there are launches from central Europe (Frankfurt and Lindenberg, Germany) and northern Europe (Sodankylä, Finland and Kiruna, Sweden) as well as a few from different sites. While the geographical distribution is therefore rather unbalanced, the flights are more widely distributed in the covered time period from January 2012 to December 2024. In panel (b) of Fig. 2 all times of AirCore starts are highlighted beside the trends in atmospheric CO<sub>2</sub> (NOAA globally averaged marine surface monthly mean data, Lan et al., 2023b).


**Figure 2.** Spatial (a) and temporal (b) distribution of all the available AirCore vertical profiles, each separated by contributing organisation (NOAA and GUF).

## 2.2 EMAC model setup






EMAC (ECHAM/MESSy Atmospheric Chemistry) is an interactively coupled state-of-the-art global chemistry-climate model. It is developed and used by a community of universities and research centres. In EMAC, the underlying dynamic atmospheric model ECHAM5 (the fifth generation of the European Centre Hamburg general circulation model, Roeckner et al., 2006) is augmented by the Modular Earth Submodel System (MESSy). This enables the integration of various submodels to describe and calculate physical and chemical processes in the entire Earth System. In this way, interactions e.g. with the land surface and the ocean as well as anthropogenic influences and feedback mechanisms can be included (Jöckel et al., 2010).

For this study we analyse results of the hindcast 'TPChange specified-dynamics reference simulation' of EMAC ('TPChange' resp. 'The Tropopause Region in a Changing Atmosphere' is a Transregional Collaborative Research Center of the German Research Foundation, <a href="https://tpchange.de/">https://tpchange.de/</a>, last access: 6 May 2025). The model setup is based on the EMAC CCMI-2022 setup (Jöckel et al., 2024), but with purely Coupled Model Intercomparison Project Phase 6 (CMIP6) boundary conditions for Ozone Depleting Substances and based on the SSP245 scenario after 2014 (Meinshausen et al., 2017, 2020). Furthermore, the model setup includes the GMXe submodel for tropospheric aerosol microphysical properties (Pringle et al., 2010), general model developments since the EMAC CCMI-2022 model simulation and some TPChange project related diagnostics. Details on the MESSy submodels and infrastructure used are listed in the supplement in Table S2 and Table S3, respectively.

The simulation covers the time period from January 2000 to June 2024, with two additional years of spin-up. EMAC is operated with a time step length of 12 minutes and with output of results (global snapshots) every 5 hours. The model has a resolution of T42L90MA: the spherical truncation of T42 corresponds to a quadratic Gaussian grid with horizontal resolution of approximately 2.8° in latitude and longitude and the hybrid sigma-pressure coordinate scheme contains 90 vertical levels up to 0.01 hPa (Jöckel et al., 2016), resulting in a vertical resolution of 500-700 m in the lower stratosphere and beneath. Newtonian relaxation ('nudging') to ERA5 meteorological reanalysis data (Hersbach et al., 2020) is performed according to Jöckel et al. (2006). Nudged are the divergence (48 hours), the vorticity (6 hours), the temperature and the logarithm of the surface pressure (both 24 hours), with the relaxation times in parentheses. The Newtonian relaxation is performed by the ECHAM routines in the spectral space. The 0th wave number, i.e. the global mean, is not nudged. The nudging process is performed mainly within the troposphere, with a maximum strength between approximately 700 hPa and 100 hPa. Therefore, the large scale stratospheric circulations (BDC) is not affected directly by nudging.

#### 2.3 EMAC CO<sub>2</sub> and artificial tracers

Alongside basic meteorological information from the simulation, we use EMAC tracer information for the species that are measured in the AirCore observations. As a simulation with the EMAC set-up-described in the previous section is a pure atmospheric model simulation without simulated sources and sinks of atmospheric tracer gases, the mole fractions at the surface are prescribed, using the submodel TNUDGE for tracer relaxation towards prescribed values (Kerkweg et al., 2006).

Due to the time resolution of the input data, these tracers directly include a seasonal cycle. The GHG mole fractions seen by other model components, such as the radiation scheme, are based on input fields from the CMIP6 protocols (Meinshausen et al., 2017, 2020). In addition to the 'standard' CO<sub>2</sub> tracer affecting radiation we included two diagnostic CO<sub>2</sub> tracers, which differ only with regard to the lower boundary conditions. Thus, using the NOAA marine boundary layer data set (NOAA MBL reference, Lan et al., 2023a) as prescribed lower boundary mole fractions in combination with two tracer relaxation vertical regions, namely (I) only at the surface to the atmosphere or (II) within the entire planetary boundary layer (PBL), results in a total of three different CO<sub>2</sub> output variables from the simulation (CO<sub>2</sub> standard, CO<sub>2</sub> MBL srf, CO<sub>2</sub> MBL pbl).





We also included an artificial deseasonalised CO<sub>2</sub> tracer (CO<sub>2</sub>\_deseas) into the simulation in order to study the seasonal cycle of CO<sub>2</sub> in a novel and unique way. This tracer is based on a deseasonalised input, which is processed by applying a package from NOAA Global Monitoring Laboratory's program for curve fitting and filtering of time series data (CCGCRV Filter, 3rd order polynomial and 4th order harmonics, <a href="https://gml.noaa.gov/ccgg/mbl/crvfit/crvfit.html">https://gml.noaa.gov/ccgg/mbl/crvfit/crvfit.html</a>, last access: 15 May 2025) to the NOAA MBL dataset to isolate the trend value. All model tracers analysed are summarised in Table 1 with their abbreviations as used in the manuscript later on.

**Table 1.** Overview of the different CO<sub>2</sub> tracers simulated with the EMAC model, with their abbreviations and details of lower boundary conditions. The EMAC-derived seasonal CO<sub>2</sub> signal is included for completeness, even if it is not a tracer per se.

| EMAC CO <sub>2</sub> tracer | lower boundary condition        |                   | Comment                                               |
|-----------------------------|---------------------------------|-------------------|-------------------------------------------------------|
| abbreviation                | Input dataset                   | tracer relaxation | -                                                     |
|                             |                                 | vertical region   |                                                       |
| CO <sub>2</sub> _standard   | CMIP6 historical GHG            | surface           | CO <sub>2</sub> as used in the framework of CMIP      |
| $CO_2\_MBL\_srf$            | NOAA MBL reference              | surface           |                                                       |
| $CO_2\_MBL\_pbl$            | NOAA MBL reference              | PBL               |                                                       |
| CO <sub>2</sub> _deseas     | Deseasonalised NOAA MBL         | PBL               | CO <sub>2</sub> like tracer which has a similar long- |
|                             | reference (using CCGCRV Filter) |                   | term trend as CO <sub>2</sub> , but no seasonal cycle |
| CO <sub>2</sub> _seas       | -                               | -                 | EMAC seasonal CO <sub>2</sub> signal calculated       |
|                             |                                 |                   | via CO2_MBL_pbl minus CO2_deseas,                     |
|                             |                                 |                   | see Eq. (1)                                           |

The combination of the various CO<sub>2</sub> outputs integrated into EMAC enables easy and accurate separation of the seasonal CO<sub>2</sub> signal (CO<sub>2</sub>\_seas), which is calculated as the difference between the CO<sub>2</sub>\_MBL\_pbl and the CO<sub>2</sub>\_deseas tracer.

$$CO_2$$
\_seas =  $CO_2$ \_MBL\_pbl -  $CO_2$ \_deseas (1)

This calculated tracer not only allows for investigation of the CO<sub>2</sub> seasonal cycle in a global and long-term perspective, but also serves as a novel diagnostic tool to improve the interpretation of simulated and observed greenhouse gas distributions with respect to aspects of atmospheric dynamics.

# 2.4 Comparison approach







The difference in resolution between the EMAC model data and the AirCore observational data requires preliminary data processing steps before a meaningful comparison can be made to evaluate the model performance. To do a point-by-point comparison, we first selected the EMAC data corresponding to the sampling time of the AirCore using the S4D submodel (Jöckel et al., 2010). Depending on the flight duration and based on the 12-minute model time step length, this results in three to five atmospheric profiles of model output, which were interpolated horizontally (bilinear) to the AirCore flight trajectory for each model pressure level (S4D submodel, Jöckel et al., 2010). As there are virtually no changes on such small time scales (< 1 h) in a global model (average variation of the EMAC profiles in a grid box within one hour was < 0.03 % for each species) we took the mean of these subsets and performed no interpolation in time. Subsequently the AirCore profile data were fitted to the vertical resolution of EMAC. Rather than using a simple linear interpolation of the vertical coordinates, a refinement of the procedure is appropriate, as there are already known challenges and uncertainties in both datasets with respect to altitude/pressure coordinates. The EMAC vertical levels are located in the centre of a grid box and are therefore representative for a certain pressure range. For AirCore profiles, the correct altitude mapping is a central challenge and target of ongoing scientific discussion (Wagenhäuser et al., 2021; Tans, 2022). Possible deviations caused by different height assignment approaches (pressure-equilibrium vs. fluid-dynamics) or bias due to uncertainties in the identification of the atmospheric sample in the AirCore (start-point determination) can add up to overall uncertainties of several hundred meters. Furthermore, mixing inside the AirCore prior to the analysis in the lab causes some blurring in the (fine)structures of the trace gas profiles. To take these aspects into account we chose a weighted average of the AirCore observations for the comparison. The weighted average was calculated for each model level based on the AirCore data in the pressure interval half the distance to the surrounding model levels.

To enable a comparison of the large number of AirCore flights, the agreement of the datasets was quantified using the mean absolute deviation (MAD) and its standard deviation ( $\sigma$ AD). Both metrics were calculated for each profile using the following equations:

$$MAD_{x,profile\ (i)} = \frac{1}{N} \sum_{j=0}^{N} \left| x_{j,EMAC} - x_{j,AirCore} \right| \tag{2}$$

$$\sigma AD_{x,profile\ (i)} = \sqrt{\frac{1}{N-1} \sum_{j=0}^{N} (\left| x_{j,EMAC} - x_{j,AirCore} \right| - MAD_{profile\ (i)})^2}$$
(3)

where i is representing the individual profile comparison, j refers to the information in the respective pressure levels of the comparison and x is the selected quantity (measured trace gas species and corresponding model tracer). This was not only done for the complete profiles, but also separately for individual pressure ranges of the profiles (including UTLS) in order to determine differences in model performance as a function of altitude.

## 3 EMAC model evaluation

## 3.1 Simulated CO<sub>2</sub> tracers







In order to properly assess the detailed comparison between EMAC and AirCore, we first briefly examine the differences between the various simulated CO<sub>2</sub> tracers to identify possible effects of the boundary conditions and nudging method used. Modelling uncertainties become visible and an indication of the impact of assumptions on the results is given.

Although the differences between the prescribed input data sets (NOAA MBL, GHG for CMIP6) are reflected in the model results, the simulated CO<sub>2</sub> distributions from both products are in good agreement globally and for the atmosphere as a whole. If at all, significant deviations exist only for short time periods and in specific areas. In the lower troposphere, there are sometimes larger deviations of a few ppm, which are more frequent and pronounced above land surfaces. Here it becomes apparent to what extent the input data includes tropospheric pollution. The GHG for CMIP6 CO<sub>2</sub> input has a slightly higher variability than the MBL reference data, which is designed to be a background low-noise representation of atmospheric CO<sub>2</sub>. However, the already weak difference of signals between the input datasets further decreases with increasing altitude. The impact is very low in the upper troposphere, with maximum deviations of less than 1 ppm, and is almost non-existent in the stratosphere.

The absolute effect of the tracer relaxation at the lower boundary (surface vs. entire PBL) on the CO<sub>2</sub> distribution is in the order of a few hundred ppb and therefore even smaller than when comparing the different input datasets. It is also reduced with height. In many cases the deviation in lower layers caused by the selected tracer relaxation vertical region seems to be compensated by an opposing discrepancy in the levels directly above. This could be an effect of the temporal delay in the CO<sub>2</sub> distribution due to combination of the extent of relaxation area and tropospheric transport processes. Because the MBL data product is derived from measurements from a subset of sites in the NOAA Global Greenhouse Gas Reference Network, where samples are predominantly of well-mixed MBL air, these values should be representative of a large volume of the atmosphere. For this reason, the use of the CO<sub>2</sub> tracer based on the MBL data in combination with relaxation throughout the entire PBL, rather than the one at the surface only, seems appropriate.

Overall, the statistics of the comparison of the three different CO<sub>2</sub> tracers with the AirCore profiles are very similar, differing only in nuances and for individual flights (see letter-value plots in Fig. S1 in the supplement for further details). Bearing in mind the impact of input and relaxation vertical regions, in the following we only use CO<sub>2</sub>\_MBL\_pbl, as it is the most globally representative one and suited best for our analysis focusing on the UTLS and the lower stratosphere. In addition, this tracer is consistent with the artificial deseasonalised CO<sub>2</sub> MBL product (CO<sub>2</sub> deseas).

## 3.2 Vertical profile comparison




Figure 3 (a-d) shows the comparison of the vertical CO<sub>2</sub> profiles derived from AirCore with the EMAC model output for four example flights from different seasons. These selected flights already reveal key characteristics of the comparison:

in most cases, in situ observations are in good agreement with the simulation results. There is no explicit systematic offset. If at all, a very small tendency for EMAC to underestimate the CO<sub>2</sub> mole fractions might be present, but certainly not exceeding a few tenths of a ppm (see statistics of the point-to-point comparison in supplementary Fig. S2). Larger structures in the CO<sub>2</sub> vertical profiles are usually similar, but occasionally slightly shifted vertically. In contrast, small-scale variations are not well-captured by the model, which was expected due to the EMAC resolution. While the deviations in the UTLS and above are generally less pronounced, the largest deviation is found in the troposphere. This can be attributed to the impact of regional CO<sub>2</sub> sources and sinks on the lower parts of the snapshot-like AirCore profiles, which can hardly be resolved in the global model using a background-like input such as NOAA MBL.

Figure 3. Comparison of the vertical CO<sub>2</sub> profiles derived from AirCore with the EMAC model output (CO<sub>2</sub>\_MBL\_pbl). Panels (a-d) show four example flights; panel (e) the statistics of the mean absolute deviation (MAD, metric for determining the similarity between simulated and observed data) per profile from all individual flights, either for the total profiles or according to subsets of specific pressure ranges representing the troposphere, UTLS and stratosphere. The boxes of the Box-Whisker-Plot extend from the first quantile to the third quantile with a line at the median. The whiskers include all data points lying within 1.5 times the interquartile range, points are outliers.

The boxplots in Fig. 3e summarize the results of the systematic comparison of all available AirCore profiles with the corresponding EMAC model profiles using the MAD metric (total number of n=182). The majority of the individual profile comparisons have a MAD in the range of 0.8 ppm to 1.8 ppm, but there are also some outliers with MADs larger than 3.5 ppm.

Focusing on the dependence of the agreement on particular pressure ranges (boxplots in the lower part of the panel) it becomes clear that the tropospheric part (> 350 hPa) is responsible for the majority of the overall deviation. Beneath 350 hPa, the distribution of MAD values is much broader and a significant amount of profiles (~15 %) shows extreme outliers with MADs partly far beyond 5 ppm, corresponding to deviations in single lower layers of up to 20 ppm. All these cases are characterised by complex and highly variable CO<sub>2</sub> AirCore profiles, whose rapid sequence of gradients in the troposphere is not fully captured by the EMAC simulation. This is reflected in the very high standard deviation of the absolute deviation, which is more than 3 ppm for these conspicuous comparisons, contrasting values between 0.5 ppm and 1.5 ppm in other cases. The profiles concerned were all taken between June and August, indicating that the strong and rapidly changing influence of the biosphere (sink) during the vegetation period is often underestimated. Therefore, these discrepancies should, at least in part, not be related to the model performance itself, but rather to the already discussed uncertainties of the input data in combination with the different resolutions of the two compared data sets.

In the stratosphere (< 50 hPa), 75 % of the profiles have a MAD of less than 1.1 ppm, indicating a highly reliable simulation of the CO<sub>2</sub> distribution by EMAC in this region. Although the MAD from the more variable region of the UTLS (50 hPa 2</sub>-rich tropospheric air into the stratosphere. Probably such a phenomenon occurs only on smaller scale, so it is captured by AirCore while the model is not able to resolve this variability. Nevertheless, EMAC reproduces the (large-scale) features of the CO<sub>2</sub> distribution in the UTLS well.

Despite the fact that a vertically weighted AirCore data approach was used for the profile comparison in order to incorporate uncertainties of height assignment, it becomes clear that a part of the deviations between model and observations is due to the linkage between trace gas mole fractions and the vertical coordinate. Thus, the absolute values and the structures in general match well, but the latter are sometimes shifted in height. These vertical shifts also appear for other simulated species such as CH<sub>4</sub> and can be seen in the temperature profiles or the position of the tropopause. Again, the coarse spatial resolution of the model is likely to be crucial. Independent of the variable, there are non-resolved small-scale and short-term variabilities that can only be seen in high-resolution AirCore observations. To evaluate the accuracy of the model simulations correctly, it should be noted that the AirCore observations used for the comparison are not globally representative.

Considering the limitations discussed, the EMAC model delivers good results. It is able to map the CO<sub>2</sub> distribution in the regions we are particularly interested in (free troposphere and above). Beyond this profile comparison, the available model output with its global extent and quasi continuous long-term time coverage, provides insights into various relationships and offers excellent opportunities for analysis that are not possible with spatially and temporally sparse observational data (alone).

## 4 Characteristics of the CO<sub>2</sub> distribution from EMAC simulation

# 4.1 Atmospheric CO<sub>2</sub> variability in the UTLS

Basic investigations of the EMAC model behaviour show that the expected general patterns of the global atmospheric CO<sub>2</sub> distribution in time and space are well reproduced (e.g. hemispheric differences in the troposphere, transport barriers, rising global trend). The simulated zonal mean distribution of CO<sub>2</sub> reveals known large-scale features of upper tropospheric and stratospheric variability. Figure 4 shows the monthly averaged cross-sections of CO<sub>2</sub>\_MBL\_pbl for even months of 2019 as an example (the same for odd months in the supplement in Fig. S3).

Figure 4. Monthly averaged zonal mean latitude-pressure cross section plots of the diagnostic EMAC CO<sub>2</sub> tracer (CO<sub>2</sub>\_MBL\_pbl) for even months of 2019. The black line is the WMO tropopause. Grey contour lines indicate jet positions (zonal windspeed; lower threshold is 20 m s<sup>-1</sup>, 5 m s<sup>-1</sup> spacing) and white lines represent selected potential temperature surfaces.

The results presented in Fig. 4 illustrate that the CO<sub>2</sub> mole fraction generally decreases with increasing altitude between 350 hPa and 10 hPa. Only in spring/early summer in each hemisphere there are, albeit weakly pronounced, (middle) stratospheric areas in polar latitudes with reversed vertical CO<sub>2</sub> gradients (e.g. highlighted by the red box in Fig. 4e). This phenomenon varies in intensity inter-annually, is more distinct in the southern hemisphere (SH) and could be associated with the polar vortices. As the feature is not easily visible in the monthly mean representation of Fig. 4, a more detailed visualisation is provided by the vertical profile plot for high latitudes shown in supplementary Fig. S4a. Compared to the other latitudes,

the tropics in the stratosphere show significantly higher mole fractions. Over the course of the year, this rather isolated sector remains, but expands as the zone with the largest meridional gradient of CO<sub>2</sub> shifts towards higher latitudes into the respective summer hemisphere. In the extra-tropics above 300 hPa and up to 50 hPa, there is a region with very clear seasonal variations (details on this are discussed later on). These main characteristics of the simulated CO<sub>2</sub> in EMAC agree with the findings of Diallo et al. (2017) for these parts of the atmosphere. Furthermore, they are in line with our understanding of the transport mechanisms of long-lived substances through the BDC (Holton et al., 1995; Butchart, 2014). Accordingly, the EMAC 'TPChange specified dynamic simulation' results as analysed in our study can be used to investigate some details of the CO<sub>2</sub> distribution more closely.

As highlighted above, there are particularly strong variations of the CO<sub>2</sub> distribution in the UTLS region. This complex, difficult-to-access transition zone of the atmosphere can be well studied in combination with the available AirCore data, and many processes there are not yet fully understood. In the respective hemispheric winter/spring poleward of 30°, the CO<sub>2</sub> mole fractions largely follow the isentropes up to about 340 K potential temperature (see Fig. 4). This extratropical part of the UTLS is delimited by the strong meridional gradient adjacent to the tropics, representing the transport barrier caused by the STJ (Miyazaki et al., 2009). During the summer months the relatively high CO<sub>2</sub> that has reached the tropical tropopause layer (TTL) and lower stratosphere via convection and upwelling expands to the mid latitude LMS (located between the local tropopause and approx. 130 hPa). This is consistent with other observations (Hoor et al., 2005; Sawa et al., 2008) and the conclusions by Bönisch et al. (2009), emphasising the stronger contribution of tropical tropospheric air in the extratropical LMS in this season due to increased quasi-horizontal tracer transport across the weaker STJ (see also Fig. S4b). Particularly in the northern hemisphere (NH), this layer of CO<sub>2</sub>-rich air in the extratropical LMS contrasts with the simultaneously decreasing tropospheric CO<sub>2</sub> due to the biospheric uptake by photosynthesis during the terrestrial vegetation period (Fig. 4d, Fig. S3e).

Asymmetries between the hemispheres regarding the CO<sub>2</sub> variability in the UTLS are visible throughout all seasons and can partly be related to the hemispheric differences in the strength and persistence of the jets. Furthermore, the shallow branch of the BDC influences the CO<sub>2</sub> distribution in the lower stratosphere. As the effects of its characteristics (e.g. seasonality, hemispheric differences) become better visible in the results of the isolated CO<sub>2</sub> seasonal signal than in the CO<sub>2</sub> simulation itself, they are discussed in section 4.4. Apart from this large-scale transport mechanism and the coupling with the tropics modulated by the strength of the STJ, the extratropical UTLS is characterised by distinct vertical gradients. This feature is particularly interesting for gaining insights into the strength of the tropopause acting as a transport barrier.

## 4.2 Components of the CO<sub>2</sub> signal





Drawing conclusions about the strength of the complex, often bidirectional exchange processes between the troposphere and the stratosphere from these trace gas gradients is not straight forward. Several components contribute to the CO<sub>2</sub> mole fraction at a given location in the atmosphere: Besides the previously discussed transport with general circulation patterns and mixing,

there is the long-term trend and the seasonal cycle. The interplay of these three signals, which are all relevant to the UTLS, makes the evaluation more difficult, as details can be masked by the signal superposition.

To give an idea of the contribution of these factors Fig. 5 visualises the temporal evolution of the monthly mean CO<sub>2</sub> from 2019 to 2023 in the 35° N to 55° N latitudinal band (corresponding to the geographic locations where most of the AirCore flights from NOAA or GUF took place). Zonal mean EMAC CO<sub>2</sub> curves for different pressure levels are shown with their standard deviation as shaded areas. Symbols represent AirCore measurements from individual flights for comparable pressure ranges. The standard deviation bars for the observations may be smaller than the symbol size. For clarity AirCore information is not added for every level.

**Figure 5.** Temporal evolution (2019-2023) of zonal mean EMAC CO<sub>2</sub>\_MBL\_pbl simulation results (lines) at different pressure levels for the 35° N to 55° N latitudinal band with their standard deviation as shaded areas. Symbols represent AirCore observations from individual flights in the vicinity of these pressure levels (mean over the pressure intervals with standard deviation as error bars).



The uncertainty ranges of the different datasets overlap for upper levels with very few exceptions. In line with the EMAC evaluation in chapter 3.2., slightly higher deviations are visible for the (lower) troposphere. Nevertheless, both, the observations and the simulation results provide a consistent picture of the  $CO_2$  variability. First of all, the rising long-term trend is striking. Regardless of the atmospheric level, the EMAC  $CO_2$  mole fraction increases with (2.38  $\pm$  0.09) ppm yr<sup>-1</sup> between 2020 and 2023, matching the annual mean global growth rates of (2.44  $\pm$  0.16) ppm yr<sup>-1</sup> reported by NOAA (Lan et al., 2023b). Focusing on pressure-dependent variability, it becomes clear that the  $CO_2$  mole fraction, on average, decreases with altitude. This tendency is superimposed by the intra-annual variability, due to the seasonal cycle. Being most pronounced near the ground, it is not only dampened in higher layers but also phase-shifted and modulated.

# 4.3 Upward propagation of the CO<sub>2</sub> seasonal cycle



To investigate the CO<sub>2</sub> seasonal cycle and its upward propagation from the troposphere across the UTLS into the LMS, we constructed a climatology of the EMAC-derived CO<sub>2</sub> seasonal signal. Therefore, we first calculated monthly means of CO<sub>2</sub>\_seas in order to subsequently compute the average behaviour per calendar month over the entire simulated time period 2000-2023. No further normalisation or detrending was necessary, since CO<sub>2</sub>\_seas, as given by Eq. (1), is already disentangled from the long-term increase. Figure 6a shows the result of the seasonal CO<sub>2</sub> signal vertical profiles for a grid cell in the mid-latitudes of the NH (similar plots for other locations in the supplementary material, Fig. S5).

**Figure 6.** (a) Climatological EMAC-derived CO<sub>2</sub> seasonal signal vertical profiles per calendar month for NH mid-latitudes. The shaded area shows the seasonal variation in WMO tropopause height. Grey lines on the right vertical axis represent selected pressure levels for which 'traditional' seasonal cycle plots of the same data are shown in (b)-(f). Note the different scales. The mean behaviour (2000-2023) is displayed together with its standard deviation (thin lines) and extremes.

The seasonal signal contribution changes in a complex way with altitude and throughout the year. As a result, the profile is reversed in terms of the sequence of lower and higher mole fractions relative to the deseasonalised CO<sub>2</sub> (Fig. 6a). In the NH mid-latitudes, the maximum of the seasonal cycle in the troposphere is in April/May, and the lowest values are found in August/September (Fig. 6a and Fig. 6f). Close to the surface the climatological range of fluctuation of the CO<sub>2</sub> seasonal signal at this location, which is graphically represented by the width of the envelope of all curves in Fig. 6a, reaches approximately

15 ppm, corresponding to a variation of roughly 3.5 % of the annual mean value caused by the seasonal cycle of CO<sub>2</sub>. Here, the regional influence of the biosphere through photosynthesis during the peak of the vegetation period in June-August is particularly noticeable due to the rapid and large change towards more negative seasonal signals and the steep gradient in the PBL. Overall, a significantly larger vertical profile variability in summer/autumn is visible. In comparison, the vertical structure of the seasonal contribution is much more uniform in the months from November to May. During these months, there is an increasingly positive contribution of the near-surface seasonal signal due to the accumulation of CO<sub>2</sub> by respiration processes. In the free troposphere (800-300 hPa) the seasonal signal is already slightly attenuated and lags somewhat behind the processes near the ground, which is nicely visible in Fig. 6a when the biosphere moves from net CO<sub>2</sub> sink to source (September/October) or vice versa (May/June), indicating propagation processes (rapid, but not instantaneous tropospheric transport and mixing).

As can be seen in Fig. 6a, the strongest modulation of the average monthly vertical CO<sub>2</sub>\_seas profiles with altitude occurs quite independently of season in the range between 300 hPa and 70 hPa, which is within the extent of the extratropical UTLS region. This change is primarily characterised by a dampening of the tropospheric variability and a shift in the seasonal cycle. Nevertheless, the seasonal signal is clearly visible up to 50 hPa (approx. 20 km), highlighting the importance of and need for such high-reaching observational data as can be provided by balloon-flights with AirCore. The peak-to-peak amplitude of the seasonal signal is still over 9 ppm at approx. 300 hPa (see Fig 7a), which is not substantially less than at 700 hPa. At altitudes above that level, the absolute effect of variation decreases sharply. At 150 hPa, it has dropped to just under 3 ppm, after which the decline slows down again. From around 40 hPa upwards, there is almost no change in amplitude anymore and the monthly averages of the CO<sub>2</sub> seasonal cycle signal have virtually converged, but a residual influence of about -0.2 ppm seems to remain.

This stratospheric residual is likely to be explained by the combination of processes in the large-scale upwelling region for trace gas transport through the BDC: the tropical tropopause layer. The tropical upward mass flux is seasonally variable, peaking in the NH winter months and being roughly only half as strong in the NH summer (Rosenlof, 1995; Randel et al., 2008; Yang et al., 2008; Seviour et al., 2012). The variation is approximately sinusoidal. While this seasonally varying mass flux has no effect on a tracer without a seasonal cycle, like e.g. the deseasonalised CO<sub>2</sub> tracer in the model, it will affect a tracer that shows a seasonal cycle in the input region, like CO<sub>2</sub>, as different months are weighted differently. The mean CO<sub>2</sub> seasonal cycle near the tropical tropopause reaches its local maximum in June/July and minimum in October/November (see Fig. S5b for EMAC results, consistent with Nakazawa et al., 1991; Andrews et al., 1999). Thus, the seasonal cycle is lagged by two months to the mid-latitudes of the NH. Moreover, it is a little asymmetric in shape. When both factors are weighted to calculate a mean CO<sub>2</sub> input into the stratosphere using only the seasonal signal, the result is clearly negative. This is consistent with the negative stratospheric residual from the EMAC simulations described above. Observations of CO<sub>2</sub> are used to investigate stratospheric transport times and the mean age of air, i.e. the average transit time for air from the entry point to the stratosphere to a given location in the stratosphere (see e.g. Garny et al., 2024). The finding of a negative stratospheric residual implies that the existence of the CO<sub>2</sub> seasonal cycle slightly delays the occurrence of a certain CO<sub>2</sub> mole fraction in the

stratosphere. The residual of -0.2 ppm is not considered when calculating mean age from observations, so this could lead to an underestimation of about one month in CO<sub>2</sub>-derived mean age, as it corresponds roughly to the long-term increase of CO<sub>2</sub> in one month at current increase rates.




Figure 7 illustrates the key characteristics of the EMAC derived seasonal signal as a function of altitude for the example location introduced in Fig. 6. Panel (d) provides a schematic representation of the quantities form the other panels in the context of a classical seasonal cycle view. Apart from the obvious change in amplitude with decreasing pressure (Fig. 7a), a distinct temporal shift of the seasonal cycle in the UT and LMS with respect to the troposphere is evident (Fig. 7b). The identified months with the largest average contributions of the seasonal signal remain mostly the same or shift only marginally from surface regions up to about 250 hPa. Above this level, the cycles are no longer in phase with the lower troposphere and there is a fast temporal shift of the extremes with decreasing pressure. At 100 hPa this leads to an inverted contribution of the CO<sub>2</sub> seasonal signal with the most positive values in late summer and the most negative ones shortly after the turn of a year, corresponding to a 4-month lag of the seasonal cycle in the LMS.

**Figure 7.** Key features of the climatological EMAC-derived CO<sub>2</sub> seasonal signal for the same location as in Fig. 6 with information on the vertical distribution of (a) peak-to-peak amplitude, (b) months of maximum and minimum and, (c) the time interval between the extremes. The illustration in (d) shows a schematic representation of the quantities form the other panels in the context of a classical seasonal cycle view.

Taking into account the uncertainties caused by different resolutions, coordinate systems and time scales, these EMAC results presented here are consistent with previous findings regarding the change of the CO<sub>2</sub> seasonal cycle in the LMS. During the SPURT aircraft project, a reduction of the peak-to-peak amplitude of the CO<sub>2</sub> seasonal cycle to about 3 ppm with a minimum in January and a maximum in August was observed for the European LMS (3-7 km above the local 2 PVU tropopause) (Gurk et al., 2008; Bönisch et al., 2008). Results from CONTRAIL along the flight route between Japan and Europe show for the 20-50 K interval over the 2 PVU tropopause a CO<sub>2</sub> peak-to-peak amplitude of 4.4 ppm with late winter to April minima and September maxima (Sawa et al., 2008). Diallo et al. (2017) found for the lower stratosphere (14-15 km) in the extra-tropics (zonal mean for 50°-60° N) with their TRACZILLA calculations a maximum in August and a minimum in February (in some years also somewhat later) and peak-to-peak amplitude of approx. 4 ppm.



Figure 7c shows that the CO<sub>2</sub> seasonal signal time intervals between minima and maxima remain quite stable up to 100 hPa (note: climatological averages, monthly resolution). It still takes four months from one minimum to the next maximum, and eight months from that maximum to the next minimum. But this time gap between the location of the extreme values also begins to change at higher levels of the atmosphere (< 100 hPa), resulting in an almost complete reversal at 50 hPa.

This is also visible in the 'traditional' seasonal signal plots (seasonal signal vs. calendar month at different pressure levels) in panels b-f of Fig. 6. So, the simulation data indicate a certain tilt of the shape of the seasonal cycle in the upper part of the extratropical UTLS. As implied by this previously described compression and stretching, the former (in tropospheric parts) asymmetric wavelike signal becomes a little more symmetric. The distortion is also manifested in a flattening and widening of the minimum compared to the maximum, whose shape is more clearly preserved. However, the effect is difficult to see due to the simultaneous dampening and shifting. Nevertheless, this type of modulation suggests that several processes are involved and could be a potential indicator e.g. of the seasonally fluctuating strength of the STJ influencing the coupling of UT and LS. Thus, processes that occur both vertically and meridionally, are reflected in the characteristics of the vertical CO<sub>2</sub>\_seas profile. It is therefore important to take a more global view on the distribution of the seasonal CO<sub>2</sub> signal before discussing implications for transport patterns.

One must bear in mind that the climatological perspective presented here reveals general relationships very well, but can mask features that are less pronounced, because they occur only occasionally or relocate quickly. Therefore Fig. 6b-f includes the standard deviation of the monthly average seasonal signal as thin lines next to the average behaviour from 2000-2023, to convey the magnitude of the inter-annual variation of the CO<sub>2</sub> seasonal cycle, which is clearly present, but does not contribute in a way that fundamentally changes the patterns mentioned before. Again, the short-term variability of the CO<sub>2</sub> distribution and of the CO<sub>2</sub> seasonal signal is definitely more complex and cannot be resolved by the global EMAC model.

# 4.4 A global perspective: The CO<sub>2</sub> seasonal signal tape recorder

The vertical distribution of the CO<sub>2</sub> seasonal signal strongly depends on latitude. Details on this in form of CO<sub>2</sub> seas vertical profile plots for selected locations from polar and mid latitudes as well as from the subtropics and tropics can be found in the different panels of Fig. S5. To provide a global overview, Fig. 8 shows the zonal mean cross-sections of the EMAC derived climatological average of the seasonal signal for odd months (even months in Fig. S6). In the troposphere, the hemispheric differences in the distribution of CO<sub>2</sub> seas are very clear, with a much stronger seasonal signal in the NH compared to the SH. This is in line with expectations and is due to the larger land coverage and therefore stronger sources and sinks in the NH. Although a hemispheric separation is visible near the equator especially in the lower parts of the troposphere (related to the boundary conditions), the NH dominates the seasonal signal in the free and upper troposphere well into the SH, with a time lag caused by transport times. The impact of the NH, where the near-surface fluctuation range of the seasonal cycle is 5-10 times higher as in the SH (see peak-to-peak amplitude plot in Fig. S7 for details), extends partly to around 40° S. Tropospheric gradients around those latitudes can be related to the circulation pattern of the Hadley cell.

Focusing on the shape of the gradients of CO<sub>2</sub>\_seas in the UTLS at about 30° N to 40° N, differences over the course of the year are prominent, again indicating an annual variation in the coupling of the tropics and extra-tropics. In March (end of NH winter) a very pronounced meridional gradient is observed implicating a distinct transport barrier. In this period, the CO<sub>2</sub>\_seas isolines in the STJ region run parallel to the tropopause. In contrast, during July (NH summer) the gradient is much weaker, and the CO<sub>2</sub>\_seas isolines intersect the tropopause. This is consistent with our understanding that the strongest exchange between tropical UT and the NH extratropical LMS occurs in the summer, when the STJ and the associated mixing barrier are weakest (Bönisch et al., 2009). Unlike the gradient across the STJ, the dispersion and fairly uniform distribution of the seasonal signal in the extratropical LMS throughout all months indicates quite fast quasi-horizontal mixing.

**Figure 8.** Zonal mean latitude-pressure cross-sections of the EMAC-derived climatological average (2000-2023) of the CO<sub>2</sub> seasonal signal for odd months. The black line indicates the WMO tropopause. See supplementary Fig. S6 for even months including contour lines indicating jet positions (zonal windspeed).

In the vicinity of the tropical tropopause (located at approximately 100 hPa,  $20^{\circ}$  S to  $20^{\circ}$  N), the EMAC derived seasonal CO<sub>2</sub> signal for an individual month (different panels of Fig. 8) is horizontally relatively homogeneous in space throughout the year, which is in line with observations (e.g. Park et al., 2007). This area is particularly interesting, because it is the main entry region to the stratosphere through tropical upwelling as part of the BDC (Butchart, 2014). Originating in the tropical tropopause region a slow vertical propagation of the CO<sub>2</sub> seasonal signal into the stratosphere can be observed in the tropical reservoir ( $20^{\circ}$  S to  $20^{\circ}$  N). This is consistent with findings on the upward advection of the seasonal cycle of other trace gases such as H<sub>2</sub>O (Mote et al., 1996; Randel et al., 2001) or CO (Schoeberl et al., 2006), referred to as the 'atmospheric tape recorder'.

The overall influence of the seasonal cycle is significantly larger and higher reaching in the tropics than at other latitudes in the stratosphere. With upward propagation, the signal is dampened by mixing processes, but the peak-to-peak amplitude in the tropics is still more than 1 ppm at 40 hPa. In the extra-tropics, a comparable strength of the CO<sub>2</sub> seasonal cycle is already reached at a pressure level of about 90 hPa and the signal slowly disappears from 30 hPa upwards, respectively. These differences in the weakening of the seasonal signal with altitude, especially the distinctiveness and traceability of CO<sub>2</sub> seasonality depending on latitude can be seen particularly well by comparing the vertical tape recorder images of CO<sub>2</sub> seas in Fig. 9. Each panel (a-e) shows a time versus pressure section of zonal mean CO<sub>2</sub> seas for different latitudes. Hemispheric

differences and inter annual variabilities are apparent at first glance. The former are mainly characterised by the significantly more pronounced CO<sub>2</sub> seasonal signal in the NH compared to the SH and the delay in the sequence of negative and positive signals by half a year between the hemispheres. Furthermore, it becomes visible that the closer we get to the poles, the lower the altitude up to which the relatively uniform contribution of the tropospheric seasonal cycle at this location is preserved, nicely accompanied by the position of the tropopause (black line in Fig. 9)

**Figure 9.** Time versus pressure sections of the zonal mean EMAC-derived CO<sub>2</sub> seasonal signal (vertical tape recorder) for different latitudes, representing the tropics as well as the mid-latitudes and high latitudes of both hemispheres. The black line is the WMO tropopause and the grey line is the 380 K potential temperature surface.

The imprint of the upward propagation of CO<sub>2</sub>\_seas in the tropics (Fig. 9a), which is traceable in these illustrations by the rightward tilting of the contour lines starting in the TTL, is very pronounced and far-reaching. The successive minima and maxima at the tropical tropopause can be tracked for more than a year rising to the middle stratosphere. Although the tape recorder vertical velocity is not exactly equivalent to the BDC transport velocity, it is quite close (Schoeberl et al., 2008), and so the ascent rate of these features can be used to approximate timescales of the upwelling process. Based on the positive CO<sub>2</sub> seasonal signal associated with the NH late summer maximum at 100 hPa we estimate that it takes the mode six months to reach 50 hPa. Transit time from the tropical tropopause to 30 hPa is nine months and to 10 hPa, 13 months. These are average

values for the period shown from 2015-2020. They agree with estimates based on the water vapour tape recorder (e.g. Mote et al., 1996; Strahan et al., 2009). The signature contains small but distinct inter annual variations, which are reflected by an acceleration or delay of the signal arrival at a certain pressure level by up to one or two months. This might be associated with effects of the quasi-biennial oscillation (QBO) or fluctuations in the tropical upwelling tied to El Nino – Southern Oscillation (ENSO) events. Moreover, the non-uniform transition from tape head to the tail suggests a seasonality in the tropical upward mass flux into the stratosphere. While the positive signal at the tropical tropopause is traveling only slightly upwards from NH spring to NH autumn (see Fig. 9a), an accelerated vertical transport (steeper slope of the isolines) takes place from October onwards. This is in line with our current understanding of the climatological structure of the tropical upwelling part of the BDC (Randel et al., 2008; Yang et al., 2008), which is stronger in NH winter.

The NH mid-latitude vertical tape recorder image (Fig. 9b) shows an initially slow ascent of the positive seasonal signal from the local tropopause (black line) starting in January, which is followed by a faster rise from late NH spring onwards. The transition to the negative signal, which dominates the source region for this flushing in late summer, is rapid and therefore this signal seems to propagate much faster upwards in the vertical tape recorder. Even if the effects of the Asian monsoon anticyclone (accelerated upwelling and mixing into the stratosphere during NH summer) might be visible in the presented zonal averages, the described features in CO<sub>2</sub>\_seas are not necessarily associated with vertical processes exclusively. Instead, the transport from the (sub)tropics is likely to be the main origin for these different propagation patterns. As the flushing of the NH extratropical LMS with tropical air and fast quasi-horizontal mixing across the STJ is larger in summer and autumn, this might also be reflected in a faster rise of CO<sub>2</sub>\_seas. The change in the propagation of the positive CO<sub>2</sub>\_seas above the 380 K potential temperature surface (grey line), which is clearly visible from May onwards, suggests that also other (meridional transport) processes are involved, influencing the distribution of the seasonal signal. As the change occurs above a level that air masses can only reach via the tropical tropopause, this feature is probably closely related to the shallow branch of the BDC. It could be due to a shift in transport regimes towards a dominant influence of BDC from that altitude upwards. Moreover, the lower branch of the BDC shows a seasonality as well, with considerably shorter residual transport timescales in summer than in winter (Bönisch et al., 2009).

Quite similar and equally complex patterns as in the NH mid-latitude vertical tape recorder are visible for the NH polar latitudes (Fig. 9d), where there additionally appears to be a bump in the upward propagation of the positive signal in late spring in some years, potentially related to inter annual variability of the strength of the polar vortex and associated downwelling. However, it becomes clear, that outside the tropics, it is difficult to draw conclusions about the upward propagation of the seasonal signal from the vertical tape recorder images, as horizontal transport processes also play a stronger role and their impacts are not easily visible in this representation. In this case, a horizontal cross-section provides a much better view of the dispersion of CO<sub>2</sub> seas.

Figure 10 shows patterns in the EMAC CO<sub>2</sub> seasonal signal as a function of latitude and time at different pressure levels, again highlighting the separation of tropics and extra-tropics in the lower stratosphere. In the UT at 330 hPa (Fig. 10f) the hemispheric differences in the CO<sub>2</sub> seasonal signal are still very present, and the tropics are dominated by the NH. With

increasing altitude, the hemispheres begin to converge with respect to the structures in the CO<sub>2</sub>\_seas distribution, so that the maximum of the seasonal signal beyond 40° at 127 hPa occurs uniformly in late NH summer (Fig. 10b). It is no longer delayed by 4-5 months in the SH relative to comparable latitudes in the NH, as is the case in the free troposphere, but only minimally. However, the signal amplitude is still different and there is a stronger meridional gradient around 30°-50° S compared to around 20°-50° N, hinting to a more pronounced separation in the SH, which could be linked to the stronger STJ in this hemisphere. Slightly above the tropical tropopause (81 hPa, Fig. 10e) the CO<sub>2</sub> seasonal signal propagates rapidly from the tropical reservoir into both hemispheres. This feature is consistent with the results by Boering et al. (1996). In analogy to the propagation of the seasonal cycle of water vapour, which occurs both vertically (Mote et al., 1996) and also horizontally (Randel and Jensen, 2013), we refer to this signal as horizonal (latitudinal) tape recorder. The transit time of the EMAC CO<sub>2</sub> seas mode to 50° at this level is about 4 months, which is close to, but tends to be slightly longer than that visible from the horizontal water vapour dispersion (dry mode, Randel and Jensen, 2013). In the middle stratosphere EMAC finally simulates an almost completely hemispheric symmetric picture and the CO<sub>2</sub> seasonal signal is not visible anymore poleward of 20° (Fig. 10a).

Figure 10. Time versus latitude sections of the zonal mean EMAC-derived  $CO_2$  seasonal signal for different pressure levels. The black horizontal lines represent the equator (solid) and the limits of the tropics (dashed,  $\pm 20^{\circ}$ )

## 5 Conclusion and Outlook








In this study, we present an evaluation of the UTLS and stratospheric CO<sub>2</sub> distribution as simulated with the global chemistry-climate model EMAC, using high resolution AirCore observational data. Based on more than 200 high-reaching AirCore vertical profiles from the NOAA AirCore database and a newly established and ongoing bi-monthly measurement series from Frankfurt (Germany), a point-to-point comparison with hindcast simulations was performed. There are no fundamental or systematic deviations. Taking into account the limitations of the comparison, especially the different resolutions and possible uncertainties in the AirCore altitude attribution, the agreement is very good with a median absolute deviation of less than 0.9 ppm. Although observed small-scale and short-term variability obviously cannot be resolved by the coarse global model, EMAC is able to well capture larger structures.

The complexity of atmospheric CO<sub>2</sub> patterns illustrate the enormous amount of information contained in the distribution of this trace gas, but also the multitude of mechanisms that contribute to it. By simulating an artificial deseasonalised CO<sub>2</sub> tracer with the EMAC model, we separate the CO<sub>2</sub> seasonal signal from long-term trend and transport-induced variability. This novel approach has allowed us to investigate the CO<sub>2</sub> seasonal cycle in a unique way in remote areas of the atmosphere and on a global scale.

The tropospheric CO<sub>2</sub> seasonal cycle propagates upwards into the lower stratosphere and is weakened by mixing processes during transport. In the NH mid-latitudes, the strongest modulation with altitude occurs in the UTLS region, characterised by a dampening of 50 % of the amplitude of the seasonal cycle and a 4-month phase shift between 300 hPa and 100 hPa. Above this, the seasonal cycle shape is also distorted, highlighting that the CO<sub>2</sub> seasonal signal is affected by the interplay of several involved processes. In the extratropical LMS it is strongly influenced by the seasonally varying coupling with the tropics, depending on the transport barrier strength of the STJ, for which the presented global distribution of CO<sub>2</sub>\_seas provides evidence. The vertical CO<sub>2</sub>\_seas profile in the SH mid-latitudes is predicted to have the largest amplitude in the LMS.

In the stratosphere, we identified both, a vertical and a horizontal tape recorder of the seasonal CO<sub>2</sub> cycle. It is linked to the seasonality of CO<sub>2</sub> in the TTL, which propagates not only upwards within the BDC upwelling, but also horizontally into both hemispheres as part of the shallow branch. Above about 50 hPa, the hemispheric asymmetry of the CO<sub>2</sub> seasonal cycle, which characterises the tropospheric and even the UTLS distribution is not visible anymore in the model results. While the seasonal signal outside the tropics vanishes at the transition to the middle stratosphere, it can be clearly traced travelling up to 10 hPa within slightly more than a year in the tropical reservoir. These modal transport time scales derived from the tape recorder imprint with EMAC are consistent with results from studies of the water vapour tape recorder (Mote et al., 1996; Strahan et al., 2009).

Although the presented implications on and aspects of atmospheric transport are largely well-known and discussed using various trace gases, the seasonal signal of CO<sub>2</sub> offers a powerful (one for all) diagnostic to investigate them, as it carries information on transport at different time scales and on mixing. Moreover, our isolation of the CO<sub>2</sub> seasonal signal and its global distribution provides a quantitative estimate for the uncertainty in Age of Air calculated via CO<sub>2</sub> in different parts of

the UTLS and stratosphere. Our findings suggest that due to the coupling of the seasonality in tropical upward mass flux and the seasonality of CO<sub>2</sub>, mean age derived from CO<sub>2</sub> could be systematically underestimated by about a month.

In all these aspects we demonstrated the wide range of possible applications and the usefulness of the newly introduced deseasonalised CO<sub>2</sub> tracer. We encourage other modellers to add such a tracer to their simulations, as we believe this metric would also be extremely suitable for model intercomparisons.




Because the contribution of the seasonal cycle to the atmospheric CO<sub>2</sub> variability in the UTLS and stratosphere contains so much information on atmospheric transport, and in order to validate the patterns shown in this work with the EMAC model, it would be a worthwhile objective to isolate the seasonal signal in observational data as well. However, such an approach is challenging. Detailed strategies for separating the seasonal signal in observations must address the problem that a CO<sub>2</sub>-independent Age of Air information would be imperative to disentangle seasonality from the combined effect from transport and long-term increase. For reliable results a good coverage and representativeness of CO<sub>2</sub> measurements is required - not only at ground level, as is achieved with global measurement networks, but also at higher altitudes. Especially for regions that cannot be reached by aircraft, AirCore provides a very cost-effective sampling option that makes it possible to regularly obtain high quality trace gas data. Thus, the high-reaching AirCore vertical profiles are in principle promising to constrain the seasonality of CO<sub>2</sub> in the UTLS and above from observations.

# Data availability

Observational for NOAA AirCore Atmospheric Sampling System **Profiles** available from data are **GUF** https://doi.org/10.15138/6AV0-MY81 (Baier et al., 2021) and for AirCore profiles from 690 https://doi.org/10.5281/zenodo.15274043 (Degen et al., 2025). The EMAC simulation results (S4D EMAC profiles and 3D time series of CO<sub>2</sub> tracers) are available from https://doi.org/10.5281/zenodo.15583480 (Menken, 2025)

## Author contributions

BCB and CS contributed the NOAA AirCore Program observational data. JD and AE operated and provided data of the GUF AirCore. PJ and JMM performed the EMAC simulations. JD lead the data analysis and prepared the manuscript with contributions from TJS and AE. All co-authors were involved in the scientific discussion and editing of the manuscript.

## Code availability

The Modular Earth Submodel System (MESSy) is being continuously further developed and applied by a consortium of institutions. The usage of MESSy and access to the source code is licenced to all affiliates of institutions who are members of the MESSy Consortium. Institutions can become a member of the MESSy Consortium by signing the MESSy Memorandum of Understanding. More information can be found on the MESSy Consortium website (<a href="http://www.messy-interface.org">http://www.messy-interface.org</a>, last access: 27 May 2025).

The Python software code for data processing and evaluation can be made available by the corresponding author upon request.

## Competing interests


At least one of the (co-)authors is a member of the editorial board of Atmospheric Chemistry and Physics.

## Acknowledgements

We acknowledge Tim Newberger, Sonja Wolter, Jack Higgs and Jianghanyang Li for their support of NOAA AirCore Program flights. We are grateful for the contribution of all researchers and technical staff who helped with the GUF AirCore flights, in particular Timo Keber. We would like to thank Peter Hoor for his valuable ideas in discussions.

## Financial support

- This research was supported by the Deutsche Forschungsgemeinschaft (DFG, German Research Foundation) collaborative research programme 'The Tropopause Region in a Changing Atmosphere' TRR 301 Project-ID 428312742 and used resources of the Deutsches Klimarechenzentrum (DKRZ) granted by its Scientific Steering Committee (WLA) under project ID bd1305. Funding to support the NOAA AirCore Program was provided in part by the NOAA Cooperative Agreement with CIRES, NA17OAR4320101, NASA grant 80NSSC18K0898, and NASA/JPL subcontract 1615988.
- This open-access publication was funded by Goethe University Frankfurt.

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
