# Peer review of "CO2 variability and seasonal cycle in the UTLS: Insights from EMAC model and AirCore observational data"

_EGUsphere, 2025_

## Referee Comment (RC2)

**Review: CO2 variability and seasonal cycle in the UTLS: Insights from EMAC model and AirCore observational data**

**Summary**

The present work by Johannes Degen et al. studies the distribution and seasonal cycle of CO2 in the upper troposphere-lower stratosphere (UTLS) based on observations as well as model results in the time range of 2000 to 2024. Observational data is obtained via in-situ AirCore measurements; the global distribution of CO2 is simulated using the Atmospheric Chemistry model EMAC using CO2 tracers, from which the seasonal signal is derived. Overall, observed and modeled CO2 mixing ratios agree very well with each other. The CO2 seasonal cycle exhibits a dampening and time lags with altitude, as well as a tilt, which is related to the subtropical transport barrier. Both a vertical and a horizontal (latitudinal) tape recorder are presented, revealing information on the seasonality of the BDC tropical upwelling and shallow branch.

**General comments**

**Relevance and Overall Quality**

CO2 is of high relevance to climate studies not only due to its impact on radiative balance, but also as an almost passive tracer, revealing information on large-scale transport through its distribution and via Age of Air. The UTLS is of special importance due to the strong radiative response to changes in its GHG composition. This study provides a unique comparison of highly resolved in-situ measurements of vertical CO2 profiles in the UTLS with global Chemistry Climate Model results, as well as an innovative approach to disentangling the seasonal cycle from trends and other variability modes using model tracers. The presented findings are closely related to the seasonal variability of the BDC and the subtropical jet, rendering this study extremely valuable for understanding global circulation and the composition of the UTLS. Overall, I find the methods, results and discussion in this work outstanding; the manuscript is also well-written and features excellent graphical representations of the data.

**Strengths**

- The comparison between model and observational data reveals valuable information on detailed vertical profiles, as well as the global distribution of CO2. Both complement each other well, and it is interesting to see how closely they match.
- Thorough comparison between different boundary conditions
- Excellent, clear and original representation of data in plots
- Comprehensive analysis of CO2 seasonal cycle depending on latitude and altitude. Interesting to see a horizontal as well as a vertical tape recorder.
- Results are thoroughly discussed and linked to dynamical processes
- The deseasonalized CO2 tracer is a novel approach to disentangling the seasonal CO2 cycle from long-term trends and interannual variability
- AirCore observations provide composition profiles that are more highly resolved than satellite measurements and reach farther into the stratosphere than aircraft measurements

**Weaknesses**

- Some parts of the discussion could be clarified further, see detailed comments below. Especially during the discussion of figures, it would be helpful to refer to individual subplots more often.
- Some of the figures require minor polishing (see "Figures" section below)

- Grammar could be refined in some places, also decide between American and British English

**Specific comments**

**Text**

- 267: Please provide a bit more detail on the statistical methods. Which correlation coefficient are used; e.g., Pearson's/Spearman's? Were the assumptions for computing correlation coefficients checked, e.g., normality? How is the Mean Absolute Deviation (MAD) computed here?
- 336-339: Did I understand the reasoning here correctly such that a weak tropopause leads to enhanced transport of CO2-rich air into the stratosphere, which is seen in AirCore observations, but not in the model, leading to the deviations between both datasets? That would seem plausible to me. If so, please clarify that in the text, since I found the connection between "weak tropopause" and "scale problem" not so obvious.
- 348: With "compared data", do you mean the AirCore observations, since they only cover specific regions? If yes, please clarify that in the text.
- 365: Is this related to the figure? If yes, it's better to start with, e.g., "Figure 4 shows...". Also, if this is part of the figure discussion, please use the same height coordinate as in the figure (between 8-35 km -> use pressure instead).
- 365: "hemispheric spring", which hemisphere is meant here? Or reword to "spring in each hemisphere". Also please refer to individual subfigures to support your point.
- 371: Could you briefly mention/discuss why this region shows enhanced seasonality?
- 384: Clearer: "contribution of tropical air in the extratropical LMS..." or "export of tropical air into the extratropical LMS..."
- 385-387: Please refer to individual subfigures for clarity. Also, which layer is meant here? Are you referring to the decrease of CO2 near the surface as seen in the 2019-08 plot? Is the main point here that, in summer, fast quasi-isentropic mixing of CO2-rich air into the extratropics counteracts the CO2 sink due to photosynthesis, or that a layer of low CO2 (biogenic) can be observed below the CO2-rich air (mixing) in the LMS? Please clarify.
- 389: Please elaborate on the hemispheric asymmetries with regards to jet strength. E.g., which hemisphere usually shows the stronger jet, and how does that influence the CO2 distribution?
- 390: Which characteristics of the shallow BDC branch (e.g., hemispheric differences and seasonal variability) can you see in your results?
- 399: ...the long-term trends and seasonal cycle of CO2 sources and sinks?
- 433-434: refer to panels, e.g., a) and f)
- 436: envelope of the curves in Figure 6a)?
- 442: Please specify in which pressure region the free troposphere lies
- 442-445 and 446-448: Please refer to individual panels; are these sentences discussing Fig. 6a?
- 455: What exactly does the stratospheric residual mean/ how can we interpret it? From Eq. 1, I understand that CO2_seas is the seasonal deviation from the deseasonalized "baseline"; so does that mean the seasonal signal of CO2 in the stratosphere is permanently negative? How exactly can we infer negative CO2 flux into the stratosphere from that? I do understand the reasoning in the following lines (455-463), but still, the meaning of the residual isn't clear to me.
- 463: By the "findings described above", do you mean the negative residual?
- 472: with decreasing pressure?
- 492: "15 km", please use altitude coordinates consistent with the figure (or add a km scale to the figure).
- 504: What do you mean by "features that are not so pronounced"?
- 513: "expected" because of the larger land coverage and therefore stronger sources and sinks in the NH? Also, it would help to explicitly describe the hemispheric differences in this sentence, i.e., stronger seasonal signal in the NH.
- 526: "...in the extratropical LMS throughout all months/seasons..."
- 532: Is this related to Fig. 8?
- 533: Homogeneous in what sense? From looking at Fig. 8, I can still see strong variations in the seasonal signal with both latitude and height.
- 544: "its distinctiveness" -> "the distinctiveness of the seasonal cycle"? Suggest rephrasing the sentence for clarity.
- 546: Suggest explicitly mentioning the hemispheric differences

- 563-566: A more detailed description of interannual variabilities would be interesting
- 568: "spring to autumn ...", "October": Please indicate that you are referring to the NH(?) and relate the observations to the corresponding subfigures.
- 569: Suggest to explicitly mention that the BDC is stronger in winter
- 571-574: I have difficulties following this part of the discussion: AMA should, as far as I know, accelerate upwelling and mixing into the stratosphere – but in NH summer. Why is the ascent starting from January linked to AMA here? Also, in "the transport from the (sub)tropics is likely to be the main origin for this feature", which feature exactly is meant here?

**Figures**

3)

- Very nice plot clearly showing the agreement between observation and model data. Subfigure 3e)
- Annotation of UTLS plot in light blue is hard to read; please choose a darker colour.
- Spell out "Mean Absolute Deviation" in the figure caption and/or mention again that this is a metric for determining the deviation/similarity between modeled and observed data (not everyone reads the Methods chapter ;)).
- A legend and additional description in the caption would help to interpret the box plots: Which data range do the coloured boxes cover, which errors are included in the error bars (only standard deviation or including systematic errors?) and what do the open circles mean (outliers?)? Do the vertical lines in the coloured boxes represent the medians?

4)

- Please add letters a)...f) to each panel.
- Since these are quite many plots: is there a specific reason why you chose to show individual months instead of seasonal averages, which might be better suited to show seasonal differences?
- In the text discussing the figure, you refer to "8-35 km", while the figure uses a pressure scale. Suggest to add a geometrical height scale to the figure, or change the discussion accordingly.
- It would help adding theta annotations to the contours in every plot
- I also recommend annotating selected wind contours with values, or at least stating the lower threshold in the figure caption. Please also specify in the caption and/or legend whether you considered zonal or horizontal wind speeds.
- Suggest rewording the caption: "...of EMAC CO2 tracer (CO2_MBL_pbl)", "...potential temperature surfaces indicating the UTLS."

5)

- Red and orange might be difficult to discern for colourblind readers
- Caption: "...these pressure levels..."
- Do the symbols represent individual AirCore flights or averages thereof?
- Otherwise, trends and seasonal cycle are very well represented here

6)

- Excellent representation of phase shifts and dampening of seasonal cycle
- Red/green colour contrast might be difficult to read for some users, suggest checking figure with a colourblind simulator or switching to explicitly colourblind-friendly palettes
- Can observations be added to this plot, or would that over-clutter it?

8)

- Add subfigure letters a)...f)
- Consistency: Why are odd months selected here, while even months are shown in Figure 4?
- Please give the time range of the climatology in the caption

**Technical corrections**

- "WMO tropopause" spells without a dash; please correct throughout the manuscript (also in the figures).
- Throughout the manuscript, "Figure" needs to be spelled out at the beginning of a sentence.

- 181/182, 202, 204, 232 etc.: Reference instead of inline link
- 211: corresponds
- 233: Spelling: "analysed" is British English, "summarized" American English
- 321: Suggest using "beneath" instead of "below" to clarify that the considered region is situated below 350 hPa in the sense of altitude, not pressure
- 324: Again, characteriZed spelled with z, use either AE or BE
- 329: no comma after "… vegetation period"
- 335: no comma after "… cases"
- 340: No return behind "… UTLS well."
- 341: Despite the fact that…
- 345: Suggest rewording for clarity: These vertical shifts also appear for other simulated species…
- 352: remove comma in "opportunities for analysis, that are"
- 381: remove comma after "CO2"
- 414, 415: Suggest writing values with uncertainties like this: (2.44 +/- 0.16) ppm/yr
- 435: climatological
- 498: simultaneous
- 500: remove comma after "both"
- 507: The "again" in "definitely more complex again" reads a little awkward here; suggest moving it to the beginning of the sentence: "Again, the short-term variability …"
- 511: Suggest rewording: "As can be seen from the panels…"
- 519: "Hadley cell" is spelled without dash
- 517, 522-524 and 541: The parts in brackets disturb the flow; suggest formulating as actual clauses
- 546: for different latitudes

---

## Author Comment (AC1)

**Response to the comments by Anonymous Referee #1**

We thank the reviewer for thorough reading of the manuscript and for providing valuable feedback. The comments and remarks have aided in improving the analysis and manuscript.

All comments are addressed in detail in the following (Reviewer's comments in black, responses in blue and quotations of the corresponding revised text passages in *italic*). The line numbers mentioned in the response refer to the revised version of the manuscript.

**General comments**

This is an original and generally well-written study (although it could perhaps be streamlined) that describes a number of interesting features pertaining to how the CO2 seasonal cycle propagates into the stratosphere. I have two major comments, one of which is practical and calls for specific, generally minor revisions to make the paper easier to follow. The other is more of an overall conceptual criticism, which doesn't necessarily need to be addressed and indeed may not be possible to address.

1) Practical. The text points out many detailed features of the figures, referencing the pressure level in hPa where they occur. Yet the figures have a sparse Y axis that is labeled only at 10^3, 10^2, and 10^1 hPa. This makes it challenging for the reader to locate the feature being described. I would suggest including more Y axis labels on the right as well as guiding lines or contours that delineate key relevant features like the STJ, the LMS, and the tropopause. This comment pertains in particular to Figures 6-8 -see also my specific comments.

We agree that a more extensive labelling of the y-axis and some additional guiding lines will improve the readability and traceability of single features in the figures. We therefore increased the tick length and adjusted the labelling to 10, 100, 1000 hPa in the figures concerned having a pressure y-axis. As suggested, we included additional y-axis labels for altitude regimes discussed in detail in the text where possible, but due to space constraints in the figures with many subplots this was not always possible. According to your tip we also introduced a box in Fig. 4 to highlight key features and added tropopause information to Fig. 6 and Fig. 7. Please note that following a suggestion from the Reviewer #2, we refer to individual subplots more frequently throughout the text in order to provide further clarification on certain aspects of the discussion.

2) Conceptual. The use of AirCore data is somewhat limited and the paper is based mainly on ECAM model output, particularly the tracer CO2_seas. This approach is justified in Figures 3 and 5, in which ECAM is shown to simulate well the observed CO2 profiles at selected latitudes (Fig 3) and the AirCore seasonal cycles at different pressure levels (Fig 5). The abstract states that CO2_seas "is a very useful diagnostic tool" but it is not clear if and how CO2_seas can be derived from AirCore observations. The authors only address this issue in the very last paragraph of the conclusions, where they admit that, "such an approach is challenging."

We understand the conceptual criticism that the aspect of calculating the CO$_2$_seas from observational data is left open in the manuscript and the discussion of the seasonal signal is mainly based on the EMAC model results. Indeed, we would have liked to include a comparison to CO$_2$_seas derived from AirCore only, but for now the seasonal cycle cannot be unambiguously separated from the combined effect of transport and long-term increase in CO$_2$ in observations. This requires further information which is not available. In particular, the lack of CO$_2$-

independent transport information for most AirCore flights inhibits the profound isolation of the seasonal $CO_2$ signal from observations, which was only enabled in the EMAC model simulations due to the specially implemented artificial tracer. Nevertheless, we are convinced that it is in principle possible to make a comprehensible and accurate calculation of a (climatological) $CO_2\_seas$ using (AirCore) measurement data. And this would be – as correctly implied by your comment – a key aspect to take the full advantage out of using the $CO_2$ seasonality as a transport tracer/diagnostic tool.

To address the concerns and to explain how $CO_2\_seas$ could be derived from observations we slightly extended the last paragraph of the conclusion.

*"[...] worthwhile objective to isolate the seasonal signal in observational data as well. However, such an approach is challenging. Detailed strategies for separating the seasonal signal in observations must address the problem that a $CO_2$-independent Age of Air information would be imperative to disentangle seasonality from the combined effect from transport and long-term increase. For reliable results a good coverage and representativeness of $CO_2$ measurements is required - not only at ground level, as is achieved with global measurement networks, but also at higher altitudes. Especially for regions that cannot be reached by aircraft, AirCore provides a very cost-effective sampling option that makes it possible to regularly obtain high quality trace gas data. Thus, the high-reaching AirCore vertical profiles are in principle promising to constrain the seasonality of $CO_2$ in the UTLS and above from observations."* (Line 678-685)

**Specific Comments**

Line 16. Please spell out EMAC (assuming ACP has a policy of no undefined acronyms in the Abstract).

Thank you, this has been changed as suggested. (Line 15)

Section 2. What is the vertical resolution of the AirCore profiles? (Later sections describing EMAC emphasize its "coarse resolution" of 90 vertical levels.)

Typical values for AirCore vertical resolution are given in Sect. 2.1 (Lines 178-179 of the original manuscript). They range from 1000 m in the lower stratosphere (~50 hPa) to less than 300 m in the UTLS and troposphere.

Please note that we only describe the overall spatial resolution of EMAC (in the sense of considering all spatial directions) in our discussion as "coarse", and not its vertical resolution alone. The horizontal EMAC resolution of approximately 2.8° in latitude and longitude primarily determines whether small-scale (vertical) features captured by AirCore can be resolved by the model. Indeed, describing the EMAC vertical resolution as "coarser" than the AirCore would not be correct for all the pressure levels discussed. In case of the EMAC set-up used in this study, the hybrid sigma-pressure coordinate scheme results in a vertical resolution of 500-700 m in the lower stratosphere and beneath. To enable an easier comparison of the vertical resolutions for the reader we slightly amended the EMAC set-up description.

*"[...] and the hybrid sigma-pressure coordinate scheme contains 90 vertical levels up to 0.01 hPa (Jöckel et al., 2016), resulting in a vertical resolution of 500-700 m in the lower stratosphere and beneath."* (Line 211)

Upon reflection, we realised that our usage of the term "resolution" in Sect 2.4 is partly misleading as we mixed up the definition of resolution as an independent data point with spatial (vertical) coverage actually intended. Accordingly, we modified the section concerned in the manuscript:

*"Subsequently the AirCore profile data were fitted to the vertical resolution of EMAC."* (Line 255)

Line 196-198 and 222-228. Please state more clearly whether the seasonal cycle of CO2 is prescribed in the standard configuration or calculated prognostically based on couple land and ocean carbon cycle modules. (Many readers will not be familiar with the details of the CMIP6 protocols.) Is the prognostic CO2 seasonal cycle from the coupled land/ocean/atmosphere model being "nudged" to the prescribed observed seasonal cycle?

The seasonal cycle of $CO_2$ is prescribed, as it is included in the input fields for the standard configuration. As the simulation is a pure atmospheric model simulation without simulated sources and sinks of $CO_2$, $CH_4$ and other atmospheric trace gases, there is no prognostic seasonal cycle from a coupled (e.g. carbon cycle) module.

For clarity we changed the general model description slightly and specified how the seasonal cycle is prescribed in section 2.3, which is focusing on details of the $CO_2$ simulation.

*"In this way, interactions e.g. with the land surface and the ocean as well as anthropogenic influences and feedback mechanisms can be included (Jöckel et al., 2010)."* (Line 198)

*"[…] As a simulation with the EMAC set-up-described in the previous section is a pure atmospheric model simulation without coupled land/ocean or carbon cycle components, the mole fractions at the surface are prescribed, using the submodel TNUDGE for tracer relaxation towards prescribed values (Kerkweg et al., 2006). Due to the monthly time resolution of the input data, these tracers directly include a seasonal cycle. The GHG mole fractions seen by other model components, such as the radiation scheme, are based on input fields from the CMIP6 protocols (Meinshausen et al. 2017, Meinshausen et al. 2020). In addition to the 'standard' $CO_2$ tracer affecting radiation we included two diagnostic $CO_2$ tracers in the simulation, which differ only with regard to the lower boundary conditions. Thus, using the NOAA marine boundary layer data set (NOAA MBL reference, Lan et al., 2023a) as prescribed lower boundary mole fractions in combination with two tracer relaxation vertical regions, namely (I) only at the surface to the atmosphere or (II) within the entire planetary boundary layer (PBL), results in a total of three different $CO_2$ output variables from the simulation ($CO_2\_standard$, $CO_2\_MBL\_srf$, $CO_2\_MBL\_pbl$)."* (Line 221-230)

Table 1 last row, last column, It would be better to describe CO2_seas as "CO2_MBL_pbl minus CO2_deseas" in the table rather than the more vague "based on CO2_MBL_pbl and CO2_deseas"?

We agree and the revised phrasing has been implemented in Table 1.

*"EMAC seasonal $CO_2$ signal calculated via $CO_2\_MBL\_pbl$ minus $CO_2\_deseas$, see Eq. 1"* (Table 1)

Line 264 change "It is calculated" to "The weighted average was calculated"

Done. (Line 264)

Line 266 What are "The two analysed species"?

We refer to $CO_2$ and $CH_4$. However, as the EMAC vs. AirCore comparison for $CH_4$ is not addressed elsewhere in the manuscript, we have decided to remove this part of the sentence to avoid confusion. Note that the entire section slightly changed, because we have added more details on the statistical methods according to a comment by Reviewer #2.

*"To enable a comparison of the large number of AirCore flights, the agreement of the datasets was quantified using the mean absolute deviation (MAD) and its standard deviation (σAD). […]"* (Line 267)

Line 312. What is meant by "on top of it"?

The phrase "on top of it" was used to mean "additionally". So, we wanted to express that (1) a global model can hardly resolve regional or local effects of $CO_2$ sources and sinks, which are visible in AirCore profiles, and (2) that this is even more the case because the model uses a background-like input (NOAA MBL).

*"While the deviations in the UTLS and above are generally less pronounced, the largest deviation is found in the troposphere. This can be attributed to the impact of regional $CO_2$ sources and sinks on the lower parts of the snapshot-like AirCore profiles, which can hardly be resolved in the global model using a background-like input such as NOAA MBL."* (Line 315-318)

Line 366. These reversed gradients are not obvious in Figure 4. Could they be shown better with vertical profile line graphs?

We agree that the reversed $CO_2$ gradients in the polar middle stratosphere are not easy to see in Fig 4. To improve this, we highlighted the concerned region in the plot with a box. Based on a suggestion from Reviewer #2 we added panel letters to the figure and slightly adjusted the associated paragraph to more precisely describe where exactly the relevant information can be found in the figure.

Thank you for your helpful suggestion to try to better illustrate this feature using a vertical profile line graph. We added such a representation of the phenomenon in the supplement (Fig. S4, panel (a), see plot in next comment) and a corresponding reference to the paragraph.

*"Only in spring/early summer in each hemisphere there are, albeit weakly pronounced, (middle) stratospheric areas in polar latitudes with reversed vertical $CO_2$ gradients (e.g. highlighted by the red box in Fig. 4e). This phenomenon varies in intensity inter-annually, is more distinct in the southern hemisphere (SH) and can be associated with the polar vortices. As the feature is not easily visible in the monthly mean representation of Fig. 4, a more detailed visualisation is provided by the vertical profile plot for high latitudes shown in supplementary Fig. S4a"* (Line 375-379)

Line 382. Similarly, this feature is not obvious in Figure 4 and perhaps could be shown in a line graph. Also, please define the approximate altitude range of the LMS.

Thank you, you are right that a range for the LMS will help the readers to follow our description and discussion of the $CO_2$ distribution results in Fig. 4. We have therefore added this to the text.

Similarly to the feature mentioned in the last comment we try to make the expansion of $CO_2$ from the tropical tropopause layer to the mid-latitude LMS and the different timescales that it takes in summer or winter clearer by preparing a line plot. This plot is added to the supplement (Fig. S4b) and is referenced in the manuscript.

[Figure]

*"During the summer months the relatively high $CO_2$, that has reached the tropical tropopause layer (TTL) and lower stratosphere via convection and upwelling expands to the mid latitude LMS (located between the local tropopause and approx. 130 hPa). This is consistent with other observations (Hoor et al., 2005; Sawa et al., 2008) and the conclusions by Bönisch et al. (2009), emphasising the stronger contribution of tropical tropospheric air in the extratropical LMS in this season due to increased quasi-horizontal tracer transport across the weaker STJ (see also Fig. S4b)."* (Line 393-397)

Line 405. By "information" do you mean "AirCore information"?

Yes, with "information" we mean "AirCore information". We changed this as suggested.

"For clarity AirCore information is not added for every level." (Line 419)

Figure 6a. The lines on the right Y axis are helpful. But why not actually label them? There would be room if the width of 6a is reduced slightly. Also, could the same labeled lines be added to Fig 7a?

Thank you for your valuable suggestions. We have adopted both (and have also included the labelled lines to the other two panels of Fig. 7).

Line 446. What exactly is meant by the "strongest modulation"?

The formulation "strongest modulation" refers to where in the average monthly vertical profiles of $CO_2\_seas$ the strongest change with altitude occurs. We changed the wording for clarity (please note that, upon reflection, we adjusted the pressure ranges in the paragraph slightly).

*"As it can be seen in Fig. 6a, the strongest modulation of the average monthly vertical $CO_2\_seas$ profiles with altitude occurs quite independently of season in the range between 300 hPa and 70 hPa, which is within the extent of the extratropical UTLS region."* (Line 462)

Additionally, we adjusted the same formulation in the conclusions:

*"The tropospheric $CO_2$ seasonal cycle propagates upwards into the lower stratosphere and is weakened by mixing processes during transport. In the NH mid-latitudes, the strongest modulation with altitude occurs in the UTLS region, characterized by a dampening of 50 % of the amplitude of the seasonal cycle and a 4-month phase shift between 300 hPa and 100 hPa."* (Line 653)

Line 448. Should 20 km be expressed in hPa, since everything else is.

Done.

*"Nevertheless, the seasonal signal is clearly visible up to 50 hPa (approx. 20 km), highlighting the importance of and need for such high-reaching observational data as can be provided by balloon-flights with AirCore."* (Line 465)

Line 454. A "residual influence of -0.2 ppm seems to remain". Is this simply CO2_seas as defined in Equation 1? Or has there been further processing of the model output? Please explain more clearly how the curves in Figure 6 are normalized/detrended to create a "climatology."

Yes, this is simply $CO_2\_seas$ as defined in Equation 1 with no further processing of the model output.

In our approach using the artificial deseasonalised $CO_2$ tracer we do not need further detrending or normalization to calculate a climatology of the seasonal signal of $CO_2$, as it is already disentangled from the long-term increase. We explained how the curves in Fig. 6 are derived in Line 422-424. For clarity we changed this description slightly.

*"To investigate the $CO_2$ seasonal cycle and its upward propagation from the troposphere across the UTLS into the LMS, we constructed a climatology of the EMAC-derived $CO_2$ seasonal signal. Therefore, we first calculated monthly means of $CO_2\_seas$, in order to subsequently compute the average behaviour per calendar month over the entire simulated time period 2000-2023. No further normalisation or detrending was necessary, since $CO_2\_seas$, as given by Eq. 1, is already disentangled from the long-term increase."* (Line 435-439)

Figure 6a and 7. Perhaps a black line showing the position of the tropopause would be useful, especially since the Y axis label has only 3 tick marks.

Thank you for your suggestion. We have incorporated the position of the tropopause to both figures (again also to the other panels of Fig.7). Please note that, since the height of the tropopause varies throughout the year, the figures show shaded areas indicating the lowest and highest monthly mean values of the WMO tropopause. We changed the caption accordingly.

*"(a) Climatological EMAC-derived $CO_2$ seasonal signal vertical profiles per calendar month for NH mid-latitudes. The shaded area shows the seasonal variation in WMO tropopause height. Grey lines on the right vertical axis represent selected pressure levels for which 'traditional' seasonal cycle plots of the same data are shown in (b)-(f). Note the different scales. The mean behaviour (2000-2023) is displayed together with its standard deviation (thin lines) and extremes."* (Caption Fig. 6)

Figure 7c. This figure is confusing. If it is not illustrating an essential point, please consider deleting.

The main reason for Fig. 7c is to show that there is a distortion of the seasonal cycle of $CO_2$, which can carry additional information on atmospheric transport. The shape of the seasonal signal is tilted above 100 hPa, which is hardly visible in the traditional seasonal signal plots in Fig. 6 b-f, due to the dampening and phase-shift that is happening simultaneously. That is why we think the representation of this tilt in this separate panel (Fig 7c.)

contains useful information, even if it might be hard to understand. We have added a pictogram to the figure to make it easier to follow what is shown in panel (c) (and in the other panels). This illustrates the quantities from Fig. 7 a-c by showing their meaning schematically in a classical seasonal cycle view. Accordingly, we made small adjustments to the figure description and caption.

[Figure]

*"Figure. 7 illustrates the key characteristics of the EMAC derived seasonal signal as a function of altitude for the example location introduced in Fig. 6. Panel (d) provides a schematic representation of the quantities form the other panels in the context of a classical seasonal cycle view. Apart from the obvious change in amplitude with decreasing pressure [...]"* (Line 487-489)

*"Key features of the climatological EMAC-derived $CO_2$ seasonal signal for the same location as in Fig. 6 with information on the vertical distribution of (a) peak-to-peak amplitude, (b) months of maximum and minimum and, (c) the time interval between the extremes. The illustration in (d) shows a schematic representation of the quantities form the other panels in the context of a classical seasonal cycle view"* (Caption Fig. 7)

Line 511.  "As can be seen"

The whole sentence was changed because of the next comment to not begin the paragraph referencing a supplementary figure.

Line 511.  Probably better not to begin the paragraph referencing a supplementary figure that most readers won't see.

Thank you for your suggestion, we changed the structure of this paragraph.

*"The vertical distribution of the $CO_2$ seasonal signal strongly depends on latitude. Details on this in form of $CO_2$_seas vertical profile plots for selected locations from polar and mid latitudes as well as from the subtropics and tropics can be found in the different panels of Fig. S5. To provide a global overview, Fig. 8 shows the zonal mean cross-sections of the EMAC derived climatological average of the seasonal signal for odd months (even months in Fig. S6)."* (Line 533-536)

Figure 8.  Can you draw in the STJ (as done in Fig 4) for March and July to illustrate the points described in Lines 520-525?  It is not obvious that the gradient is stronger in March.

From our perspective the gradient in $CO_2$_seas becomes too difficult to distinguish in doing so. To demonstrate this, we included a version of the figure containing the STJ contours (zonal wind isolines).

[Figure]

Nevertheless, we adjusted the related paragraph slightly to be more precise.

*"Focusing on the shape of the gradients of $CO_2$_seas in the UTLS at about 30° N to 40° N, differences over the course of the year are prominent, again indicating an annual variation in the coupling of the tropics and extratropics. In March (end of NH winter) a very pronounced meridional gradient is observed implicating a distinct transport barrier. In this period, the $CO_2$_seas isolines in the STJ region run parallel to the tropopause. In contrast, during July (NH summer) the gradient is much weaker, and the $CO_2$_seas isolines intersect the tropopause."* (Line 544-548)

Furthermore, we have amended Fig. S6 (the even-month version corresponding to Fig. 8) to include zonal windspeed isolines and added a reference to this in the figure caption.

*"Zonal mean latitude-pressure cross-sections of the EMAC-derived climatological average (2000-2023) of the $CO_2$ seasonal signal for odd months. The black line indicates the WMO- tropopause. See supplementary Fig. S6 for even months including contour lines indicating jet positions (zonal windspeed)."* (Caption Fig. 8)

Line 541. Is 5hPa even shown on Figure 8? If not, maybe delete this sentence.

Thank you for the hint. You are right. This feature cannot be seen in the figure (this was only true for an older version of the plot). So, we deleted this part of the sentence.

Figure 10. I am not an expert on "tape recorder" effects, but it seems like it might be a stretch to call Figure 10 a "horizontal tape recorder." Is there a precedent for this in the literature with H2O or other trace gases? Can the authors really be sure of what causes the hemispherical symmetry at and above 127 hPa? For the tropical signal to mix equally into both hemispheres seems at odds with the Brewer Dobson Circulation, which upwells in the tropics and descends into the winter hemisphere.

In analogy to the well-known water vapour vertical tape recorder (Mote et al., 1996) we refer to the dispersion of the $CO_2$ seasonal cycle originating in the TTL as a vertical and horizontal (to be more precise latitudinal) tape recorder.

The horizontal dispersion of the water vapour imprint is presented in detail in Randel and Jensen (2013). As the term "horizontal tape recorder" was not actually used in the publication by Randel and Jensen, we changed the text to state that we refer to this picture as horizontal tape recorder, as it shows a similar pattern, i.e. a seasonal signal imprinted at the tropical tropopause which then propagates and disperses in the stratosphere. In order to make this clearer we have eliminated the term horizontal tape recorder from the caption to Fig 10 and from Line 590 and added the explanatory phrase to the main text in Lines 599-602 in the original manuscript:

*"In this case, a horizontal cross-section provides a much better view of the dispersion of $CO_2$_seas."* (Line 617)

*"Slightly above the tropical tropopause (81 hPa, Fig. 10e) the $CO_2$ seasonal signal propagates rapidly from the tropical reservoir into both hemispheres. This feature is consistent with the results by Boering et al. (1996). In analogy to the propagation of the seasonal cycle of water vapour, which occurs both vertically (Mote et al., 1996) and also horizontally (Randel and Jensen, 2013), we refer to this signal as horizonal (latitudinal) tape recorder. The transit time of the EMAC $CO_2$_seas mode to 50° at this level is about 4 months, which is close to, but tends to be slightly longer than that visible from the horizontal water vapour dispersion (dry mode, Randel and Jensen, 2013)."* (Line 627-632)

With respect to the hemispheric symmetry mentioned by the reviewer which would be in contrast to the Brewer Dobson circulation (BDC), we refer (i) to the observation of the very similar signal in water vapour (Randel and Jensen, 2013) and (ii) to the fact that this is driven by quasi-horizontal transport which is expected to occur during all seasons.

Lines 632-634. I don't follow this argument. Is it important enough to be in the conclusions? In general, the conclusions should probably be trimmed to focus on the most key points.

We think it is important to keep the sentence "While the seasonal signal outside the tropics vanishes at the transition to the middle stratosphere, it can be clearly traced travelling up to 10 hPa within slightly more than a year in the tropical reservoir." in the conclusions, as it corroborates the finding of isolated tropics and mid-latitudes in the lower stratosphere.

**References**

Mote, P. W., Rosenlof, K. H., McIntyre, M. E., Carr, E. S., Gille, J. C., Holton, J. R., Kinnersley, J. S., Pumphrey, H. C., Russell III, J. M., and Waters, J. W.: An atmospheric tape recorder: The imprint of tropical tropopause temperatures on stratospheric water vapor, Journal of Geophysical Research: Atmospheres, 101, 3989–4006, https://doi.org/10.1029/95JD03422, 1996.

Randel, W. J. and Jensen, E. J.: Physical processes in the tropical tropopause layer and their roles in a changing climate, Nature Geosci, 6, 169–176, https://doi.org/10.1038/ngeo1733, 2013.

---

## Author Comment (AC2)

**Response to the comments by Anonymous Referee #2**

We thank the reviewer for thorough reading of the manuscript and for providing valuable feedback. The comments and remarks have aided in improving the analysis and manuscript.

All comments are addressed in detail in the following (Reviewer's comments in black, responses in blue and quotations of the corresponding revised text passages in *italic*) The line numbers mentioned in the response refer to the revised version of the manuscript.

**General comments**

**## Weaknesses**

- Some parts of the discussion could be clarified further, see detailed comments below. Especially during the discussion of figures, it would be helpful to refer to individual subplots more often.
- Some of the figures require minor polishing (see "Figures" section below)
- Grammar could be refined in some places, also decide between American and British English

Thank you for the general comments on how to improve the manuscript. We understand that closer connections to the text, i.e. referring to individual subplots more often, would help the discussion of figures. We have incorporated such references in several parts of the results sections and tried to clarify the parts of the discussion mentioned in the specific comments (see response to them for details). The very helpful suggestions regarding individual figures (addressed in the Figure section in detail) have certainly improved the graphical presentation of our results (e.g. adjusted colour-blindness-friendly colour scheme in Fig. 6, extended information in legends and captions). Thank you for the feedback on Grammar in the technical corrections section.

**Specific comments**

**## Text**

Line 267: Please provide a bit more detail on the statistical methods. Which correlation coefficient are used; e.g., Pearson's/Spearman's? Were the assumptions for computing correlation coefficients checked, e.g., normality? How is the Mean Absolute Deviation (MAD) computed here?

We initially calculated Spearman's correlation coefficient, as normality was not given for every comparison (Shapiro-Wilk-test to test for this; especially for the profiles where the AirCore reached not that high, the population after down-sampling to EMAC levels was sometimes quite small: < 50 per flight). During the analysis, it became clear that correlation is not necessarily a good indicator of the similarity between AirCore and EMAC CO$_2$. We found that even in cases with significant/visible deviations the coefficient was often quite high due to the "underlying" vertical CO$_2$ gradient that is in both measurements and simulations. So, in this application correlation does not tell us too much about the individual match and is quite prone to misinterpretation. For these reasons we used the other metrics (MAD and σAD). We realized while answering this question and providing additional information on the statistical methods that the correlation coefficient is not used in the results anymore. So, we

delete the listing of the correlation coefficient in the sentence on statistical methods as it is not relevant for the manuscript anymore and we simply missed to remove it previously.

We added the equations we used to calculate the MAD and its standard deviation to the manuscript and extended their description in the text.

*"To enable a comparison of the large number of AirCore flights, the agreement of the datasets was quantified using the mean absolute deviation (MAD) and its standard deviation (σAD). Both metrics were calculated for each profile using the following equations:*

$$MAD_{x,profile\ (i)} = \frac{1}{N}\sum_{j=0}^{N} \left|x_{j,EMAC} - x_{j,AirCore}\right| \qquad (2)$$

$$\sigma AD_{x,profile\ (i)} = \sqrt{\frac{1}{N-1}\sum_{j=0}^{N}\left(\left|x_{j,EMAC} - x_{j,AirCore}\right| - MAD_{profile\ (i)}\right)^2} \qquad (3)$$

*where i is representing the individual profile comparison, j refers to the information in the respective pressure levels of the comparison and x is the selected quantity (measured trace gas species and corresponding model tracer). This was done not only for the complete profiles but also separately for individual pressure ranges of the profiles (including UTLS) in order to determine differences in model performance as a function of altitude."* (Line 267-275)

Line 336-339: Did I understand the reasoning here correctly such that a weak tropopause leads to enhanced transport of CO2-rich air into the stratosphere, which is seen in AirCore observations, but not in the model, leading to the deviations between both datasets? That would seem plausible to me. If so, please clarify that in the text, since I found the connection between "weak tropopause" and "scale problem" not so obvious.

Thank you for your suggestion. Yes, this is exactly the reasoning we had in mind, and we agree that this needs to be made clearer in the text. We have amended the paragraph accordingly.

*"[...]. However, for most of these exceptions, the accompanying meteorological data from the AirCore flights indicate a rather unstable atmospheric stratification during sampling. So, a weak transport barrier at the tropopause might have led to enhanced transport of CO$_2$-rich tropospheric air into the stratosphere. Probably such a phenomenon occurs only on smaller scale, so it is captured by AirCore while the model is not able to resolve this variability. Nevertheless, EMAC reproduces the (large-scale) features of the CO$_2$ distribution in the UTLS well."* (Line 344-348)

Line 348: With "compared data", do you mean the AirCore observations, since they only cover specific regions? If yes, please clarify that in the text.

Yes, we refer to AirCore data and changed this as suggested.

*"To evaluate the accuracy of the model simulations correctly, it should be noted that the AirCore observations used for the comparison are not globally representative."* (Line 355)

Line 365: Is this related to the figure? If yes, it's better to start with, e.g., "Figure 4 shows...". Also, if this is part of the figure discussion, please use the same height coordinate as in the figure (between 8-35 km -> use pressure instead).

*This part of the text relates to the results presented in Fig. 4. We therefore adopt your suggestion of starting the description with the figure name and using the same height coordinate as in the figure.*

*"The results presented in Fig. 4 illustrate that the $CO_2$ mole fraction generally decreases with increasing altitude between 350 hPa and 10 hPa."* (Line 374)

Line 365: "hemispheric spring", which hemisphere is meant here? Or reword to "spring in each hemisphere". Also please refer to individual subfigures to support your point.

*Reworded to "spring/early summer in each hemisphere". We added panel labels to Fig. 4 and refer to them, where appropriate. Following a suggestion by Reviewer #1 we also included a box to highlight the concerned region of the reversed vertical gradient in the figure and prepared a supplementary figure with an alternative representation of this feature.*

*"Only in spring/early summer in each hemisphere there are, albeit weakly pronounced, (middle) stratospheric areas in polar latitudes with reversed vertical $CO_2$ gradients (e.g. highlighted by the red box in Fig. 4e). This phenomenon varies in intensity inter-annually, is more distinct in the southern hemisphere (SH) and can be associated with the polar vortices. As the feature is not easily visible in the monthly mean representation of Fig. 4, a more detailed visualisation is provided by the vertical profile plot for high latitudes shown in supplementary Fig. S4a"* (Line 375-379)

Line 371: Could you briefly mention/discuss why this region shows enhanced seasonality?

*As the whole next paragraph (Line 376-387) deals with the seasonality of $CO_2$ in the extratropical UTLS (probably related to the variability of the transport barrier strength of the subtropical jet) we would not like to repeat this information here. We mention this feature of the global $CO_2$ distribution at this point in the text, because it is a key characteristic and to underline that the EMAC simulation agrees with our common knowledge and previous findings in literature.*

*To make this clearer, we include a reference:*

*"In the extra-tropics above 300 hPa and up to 50 hPa, there is a region with very clear seasonal variations (details on this are discussed later on). These main characteristics of the simulated $CO_2$ in EMAC agree with [...]"* (Line 382)

Line 384: Clearer: "contribution of tropical air in the extratropical LMS..." or "export of tropical air into the extratropical LMS..."

*Thank you, changed as suggested.*

*"This is consistent with other observations (Hoor et al., 2005; Sawa et al., 2008) and the conclusions by Bönisch et al. (2009), emphasising the stronger contribution of tropical tropospheric air in the extratropical LMS in this season due to increased quasi-horizontal tracer transport across the weaker STJ (see also Fig. S4b)."* (Line 395)

Line 385-387: Please refer to individual subfigures for clarity. Also, which layer is meant here? Are you referring to the decrease of CO2 near the surface as seen in the 2019-08 plot? Is the main point here that, in summer, fast quasi-isentropic mixing of CO2-rich air into the extratropics counteracts the CO2 sink due to photosynthesis, or that a layer of low CO2 (biogenic) can be observed below the CO2-rich air (mixing) in the LMS? Please clarify.

We incorporated a reference to Fig, 4d (2019-08 plot) as we refer to the tropospheric layer of low $CO_2$ in NH summer, that is caused by biogenic uptake. This layer is not limited to the surface as even at 300 hPa the mole fraction is significantly reduced in comparison to Fig. 4c (2019-03). The main point here is that we observe such a low $CO_2$ layer below the $CO_2$-rich air in the LMS (which itself is likely due to quasi-isentropic mixing of $CO_2$-rich TTL air into the extra tropics). This might be even better visible in Fig S3e (2019-09 plot), as the tropospheric $CO_2$ minimum becomes more pronounced towards the end of the vegetation period due to the accumulative effect. Upon reflection, the statement on fast quasi-isentropic mixing in the LMS in the following sentence lacks contact to the rest of the paragraph and therefore may cause some confusion as well. As it is discussed later on (Line 525 of the original manuscript) we deleted it here.

*"[...] Particularly in the northern hemisphere (NH), this layer of $CO_2$-rich air in the extratropical LMS contrasts with the simultaneously decreasing tropospheric $CO_2$ due to the biospheric uptake by photosynthesis during the terrestrial vegetation period (Fig. 4d, Fig. S3e)."* (Line 397)

Line 389: Please elaborate on the hemispheric asymmetries with regards to jet strength. E.g., which hemisphere usually shows the stronger jet, and how does that influence the CO2 distribution?

Thank you for this feedback. We think a detailed discussion of quantitative aspects of the hemispheric asymmetries is beyond the scope of this section, which focuses on demonstrating qualitative features of the global $CO_2$ distribution. This is even more the case as we have AirCore measurements available almost exclusively in the NH.

*"Asymmetries between the hemispheres regarding the $CO_2$ variability in the UTLS are visible throughout all seasons and can partly be related to the hemispheric differences in the strength and persistence of the jets."* (Line 401)

Line 390: Which characteristics of the shallow BDC branch (e.g., hemispheric differences and seasonal variability) can you see in your results?

Details of the characteristics of the shallow BDC branch are better visible in the results of the isolated $CO_2$ seasonal signal than in the $CO_2$ simulation itself. They are discussed in section 4.4. in more detail (e.g. Line 581-583 in the original manuscript, Fig 9+10). We included a reference for this.

*"Furthermore, the shallow branch of the BDC influences the $CO_2$ distribution in the lower stratosphere. As the effects of its characteristics (e.g. seasonality, hemispheric differences) become better visible in the results of the isolated $CO_2$ seasonal signal than in the $CO_2$ simulation itself, they are discussed in section 4.4."* (Line 403)

Line 399: ...the long-term trends and seasonal cycle of CO2 sources and sinks?

Although the seasonality of the $CO_2$ sources and sinks imprints the seasonal cycle near the ground, the atmospheric $CO_2$ observations contain only their combined effects (net effect). As we mainly focus in our analyses on the $CO_2$ seasonal cycle far away from the ground (UTLS and above), we do not think that the seasonality of sources and sinks should be mentioned here.

Line 433-434: refer to panels, e.g., a) and f)

Done.

*"As a result, the profile is reversed in terms of the sequence of lower and higher mole fractions relative to the deseasonalised $CO_2$ (Fig. 6a). In the NH mid-latitudes, the maximum of the seasonal cycle in the troposphere is in April/May, and the lowest values are found in August/September (Fig. 6a and Fig. 6f).* (Line 447-450)

Line 436: envelope of the curves in Figure 6a)?

This addition in brackets was just to highlight where the range of fluctuation of the seasonal signal can be seen in Fig. 6a. We changed it slightly to be more precise.

*"Close to the surface the climatological range of fluctuation of the $CO_2$ seasonal signal at this location, which is graphically represented by the width of the envelope of all the curves in Fig. 6a, reaches approximately 15 ppm, [...]"* (Line 450)

Line 442: Please specify in which pressure region the free troposphere lies

We refer to the free troposphere (in the context of Fig. 6) as approximately 800-300 hPa. This is the pressure range (for mid-latitudes) which is throughout the year very likely not above the tropopause (see newly added shaded area representing the range of the monthly mean tropopause heights, as suggested by Reviewer #1) and rarely within the PBL (which varies in heigh form a few hundred meters above ground or even less in nighttime to up to 2 km in thermal turbulent conditions in the afternoon in summer, see Engeln and Teixeira, 2013).

For clarity, we added this pressure range to the sentence.

*"In the free troposphere (800-300 hPa), the seasonal signal is already slightly attenuated and lags somewhat behind the processes near the ground, [...]"* (Line 458)

Line 442-445 and 446-448: Please refer to individual panels; are these sentences discussing Fig. 6a?

Yes, all these sentences discuss the results shown in Fig. 6a. References to the figure have been added, where appropriate.

*"[...] which is nicely visible in Fig. 6a when the biosphere moves from net $CO_2$ sink to source (September/October) or vice versa (May/June), indicating propagation processes (rapid, but not instantaneous tropospheric transport and mixing)."* (Line 459)

*"As it can be seen in Fig. 6a, the strongest modulation of the average monthly vertical $CO_{2\_seas}$ profiles with altitude occurs quite independently of season in the range between 300 hPa and 70 hPa, which is within the extent of the extratropical UTLS region."* (Line 462)

Line 455: What exactly does the stratospheric residual mean/ how can we interpret it? From Eq. 1, I understand that CO2_seas is the seasonal deviation from the deseasonalized "baseline"; so does that mean the seasonal signal of CO2 in the stratosphere is permanently negative? How exactly can we infer negative CO2 flux into the stratosphere from that? I do understand the reasoning in the following lines (455-463), but still, the meaning of the residual isn't clear to me.

Yes, our findings from the EMAC simulation suggest that the $CO_2$ seasonal signal, as given by Eq. 1, is permanently slightly negative in the stratosphere. This stratospheric residual is explained by the input of airmasses into the stratosphere. It is not a negative flux into the stratosphere. While for the deseasonalised $CO_2$ the seasonal

variation of airmass flux to the stratosphere does not impact the average input, this is not the case for the $CO_2$ tracer with seasonal cycle. In this latter case, the average input will differ if the flux into the stratosphere occurs mainly during months where $CO_2$ is below (or above) the seasonal average. This is explained in the paragraph starting on Line 455. To make this clearer we have changed the text to now read as follows:

*"[…]. The variation is approximately sinusoidal. While this seasonally varying mass flux has no effect on a tracer without a seasonal cycle, like e.g. the deseasonalised $CO_2$ tracer in the model, it will affect a tracer that shows a seasonal cycle in the input region, like $CO_2$, as different months are weighted differently. The mean $CO_2$ seasonal cycle […]"* (Line 474)

The observation of the negative stratospheric residual implies that the existence of the $CO_2$ seasonal cycle delays the occurrence of a certain $CO_2$ mole fraction in the stratosphere by one month (0.2 ppm residual/2.5 ppm increase per year $\approx$ 1 month), without meaning that transport changed or is slower. Therefore, the findings regarding the stratospheric $CO_2\_seas$ residual might be important for studies using $CO_2$-derived AoA in the stratosphere. The transport is faster than suggested by the $CO_2$ mole fraction. We changed the explanation of the meaning of the residual in the text slightly:

*"The finding of a negative stratospheric residual implies that the existence of the $CO_2$ seasonal cycle slightly delays the occurrence of a certain $CO_2$ mole fraction in the stratosphere. The residual of -0.2 ppm is not considered when calculating mean age from observations, so this could lead to an underestimation of about one month in $CO_2$-derived mean age, as it corresponds roughly to the long-term increase of $CO_2$ in one month at current increase rates."* (Line 483-486)

Line 463: By the "findings described above", do you mean the negative residual?

Yes. For clarity we re-phrased this sentence.

*"When both factors are weighted to calculate a mean $CO_2$ input into the stratosphere using only the seasonal signal, the result is clearly negative. This is consistent with the negative stratospheric residual from the EMAC simulations described above."* (Line 478)

Line 472: with decreasing pressure?

Of course, "with decreasing pressure" is correct.

*"Above this level, the cycles are no longer in phase with the lower troposphere and there is a fast temporal shift of the extremes with decreasing pressure."* (Line 493)

Line 492: "15 km", please use altitude coordinates consistent with the figure (or add a km scale to the figure).

We decided to consistently stay with pressure for the EMAC model results altitude coordinate.

*"But this time gap between the location of the extreme values also begins to change at higher levels of the atmosphere (< 100 hPa), resulting in an almost complete reversal at 50 hPa."* (Line 512)

Line 504: What do you mean by "features that are not so pronounced"?

With this wording, we wanted to point to features that occur briefly or occasionally, or that change quickly in terms of location, so that they may not or hardly be visible in a climatological perspective, as it is presented in Fig. 6a.

For example, this applies to the effects of inter-annual circulation variability e.g. due to ENSO or QBO, that are smeared out looking only at a climatological average of CO2_seas. This is why we at least put the standard deviation of the monthly mean to the seasonal signal plots (panels b-f). We changed the text slightly to be more precise.

*"One must bear in mind that the climatological perspective presented here reveals general relationships very well, but can mask features that are less pronounced, because they occur only occasionally or relocate quickly."* (Line 524)

Line 513: "expected" because of the larger land coverage and therefore stronger sources and sinks in the NH? Also, it would help to explicitly describe the hemispheric differences in this sentence, i.e., stronger seasonal signal in the NH.

Thank you for this hint. We included both the explanation for the "expected differences" and explicitly named how they look like.

*"In the troposphere, the hemispheric differences in the distribution of CO2_seas are very clear, with a much stronger seasonal signal in the NH compared to the SH. This is in line with expectations and is due to the larger land coverage and therefore stronger sources and sinks in the NH."* (Line 536)

Line 526: "...in the extratropical LMS throughout all months/seasons..."

Changed as suggested.

*"Unlike the gradient across the STJ, the dispersion and fairly uniform distribution of the seasonal signal in the extratropical LMS throughout all months indicates quite fast quasi-horizontal mixing."* (Line 550)

Line 532: Is this related to Fig. 8?

Yes, this description is related to Fig. 8. We included a reference to this figure to clarify (see next comment for details as it deals with the same paragraph)

Line 533: Homogeneous in what sense? From looking at Fig. 8, I can still see strong variations in the seasonal signal with both latitude and height.

Thank you for your feedback. You are right, this is confusing. Of course, there are strong variations of CO2_seas in the tropical tropopause region throughout the year. We wanted to emphasize that in each individual month (different panels of Fig. 8) in the vicinity of the tropical tropopause (130-80 hPa) the variation with altitude is quite the same for the whole latitudinal range of the tropical reservoir (20° S – 20° N). So, the CO2_seas distribution in this region is horizontally quite homogenous in space.

*"In the vicinity of the tropical tropopause (located at approximately 100 hPa, 20° S to 20° N), the EMAC derived seasonal CO2 signal for an individual month (different panels of Fig. 8) is horizontally relatively homogeneous in space, which is in line with observations (e.g. Park et al., 2007)."* (Line 557)

Line 544: "its distinctiveness" -> "the distinctiveness of the seasonal cycle"? Suggest rephrasing the sentence for clarity.

Thank you, changed as suggested.

*"These differences in the weakening of the seasonal signal with altitude, especially the distinctiveness and traceability of $CO_2$ seasonality depending on latitude, can be seen particularly well by comparing the vertical tape recorder images of $CO_2$_seas in Fig. 9."* (Line 568)

Line 546: Suggest explicitly mentioning the hemispheric differences

We have expanded the paragraph to include a brief description of the two main hemispheric differences.

*"[...] Hemispheric differences and inter annual variabilities are apparent at first glance. The former are mainly characterized by the significantly more pronounced $CO_2$ seasonal signal in the NH compared to the SH and the delay in the sequence of negative and positive signals by half a year between the hemispheres."* (Line 571)

Line 563-566: A more detailed description of interannual variabilities would be interesting

Thank you for your feedback. We agree that the interannual variation of the $CO_2$_seas upward propagation is an interesting feature. Particularly because such variations might be associated with effects of the quasi-biennial oscillation (QBO) or fluctuations in the tropical upwelling tied to El Nino – Southern Oscillation (ENSO) events, a more detailed discussion of the variabilities would inevitably require a greater amount of background information on these circulation phenomena. We think this topic is beyond the scope and would rather distract from the section's overall focus on obtaining a global overview of the $CO_2$_seas distribution.

Line 568: "spring to autumn ...", "October": Please indicate that you are referring to the NH(?) and relate the observations to the corresponding subfigures.

We modified the text slightly by including the hemispheric seasons and again refer to the subfigures more often.

*"Based on the positive $CO_2$ seasonal signal associated with the NH late summer maximum at 100 hPa we estimate [...]"* (Line 585)

*"While the positive signal at the tropical tropopause is traveling only slightly upwards from NH spring to NH autumn (see Fig. 9a), an accelerated vertical transport (steeper slope of the isolines) takes place from October onwards."* (Line 593)

Line 569: Suggest to explicitly mention that the BDC is stronger in winter

Changes as suggested.

*"This is in line with our current understanding of the climatological structure of the tropical upwelling part of the BDC (Randel et al., 2008; Yang et al., 2008), which is stronger in NH winter."* (Line 595)

Line 571-574: I have difficulties following this part of the discussion: AMA should, as far as I know, accelerate upwelling and mixing into the stratosphere -- but in NH summer. Why is the ascent starting from January linked to AMA here? Also, in "the transport from the (sub)tropics is likely to be the main origin for this feature", which feature exactly is meant here?

Thank you for your feedback. You are right, AMA should accelerate upwelling and mixing into the stratosphere in NH summer and not in NH winter. We want to state that in principle effects of the Asian monsoon anticyclone might contribute to the $CO_2$_seas/be visible in the vertical tape recorder image for the NH mid-latitudes (Fig. 9b), as we calculated zonal averages. This was not meant to be related to the description of the ascent of the positive

seasonal signal in the first sentence of the paragraph, but more to be a general statement (which is - as you mentioned - most relevant for NH summer).

In the sentence stating that "the transport from the (sub)tropics is likely to be the main origin for this feature", we are referring to the differences in $CO_2\_seas$ propagation patterns, more specifically the faster rise of the negative seasonal signal in NH summer compared to the slower rise of the signal in other seasons (i.e. the positive signal in winter, or even the negative signal in late autumn).

So, we changed the sequence of sentences in this paragraph to clear up the confusion and adjusted the text to be more precise.

*"The NH mid-latitude vertical tape recorder image (Fig. 9b) shows an initially slow ascent of the positive seasonal signal from the local tropopause (black line) starting in January, which is followed by a faster rise from late NH spring onwards. The transition to the negative signal, which dominates the source region for this flushing in NH summer, is rapid and therefore this signal seems to propagate much faster upwards in the vertical tape recorder. Even if the effects of the Asian monsoon anticyclone (accelerated upwelling and mixing into the stratosphere during NH summer) might be visible in the presented zonal averages, the described features in $CO_2\_seas$ are not necessarily associated with vertical processes exclusively. Instead, the transport from the (sub)tropics is likely to be the main origin for these different propagation patterns. As the flushing of the NH extratropical LMS with tropical air and fast quasi-horizontal mixing across the STJ is larger in summer and autumn, this might also be reflected in a faster rise of $CO_2\_seas$. The change in the propagation of the positive $CO_2\_seas$ above the 380 K potential temperature surface (grey line), which is clearly visible from May onwards, suggests that also other (meridional transport) processes are involved, influencing the distribution of the seasonal signal."* (Line 597-607)

**Figures**

Thank you for your detailed feedback on individual figures or panels. We adopted most of the suggestions and this aided in improving the graphical representation of the results.

Figure 3. Very nice plot clearly showing the agreement between observation and model data.

Subfigure 3e)

- Annotation of UTLS plot in light blue is hard to read; please choose a darker colour.
  Done.
- Spell out "Mean Absolute Deviation" in the figure caption and/or mention again that this is a metric for determining the deviation/similarity between modeled and observed data (not everyone reads the Methods chapter ;)).
  Done. See changed version of the caption in the response to the next comment.
- A legend and additional description in the caption would help to interpret the box plots: Which data range do the coloured boxes cover, which errors are included in the error bars (only standard deviation or including systematic errors?) and what do the open circles mean (outliers?)? Do the vertical lines in the coloured boxes represent the medians?

The data range for the coloured boxplots from the different pressure subsections is already given in the y-label. The representation is a commonly used 'standard' Box-Whisker-Plot. So, we do not add a legend, but as suggested we put information on its different parts to the caption.

*"Comparison of the vertical $CO_2$ profiles derived from AirCore with the EMAC model output ($CO_2\_MBL\_pbl$). Panels (a-d) show four example flights; panel (e) the statistics of the mean absolute deviation (MAD, metric for determining the similarity between simulated and observed data) per profile from all individual flights, either for the total profiles or according to subsets of specific pressure ranges representing the troposphere, UTLS and stratosphere. The boxes of the Box-Whisker-Plot extend from the first quantile to the third quantile with a line at the median. The whiskers include all data points lying within 1.5 times the interquartile range, points are outliers."* (Caption Fig. 3)

Figure 4.

- Please add letters a)...f) to each panel.
  Done.

- Since these are quite many plots: is there a specific reason why you chose to show individual months instead of seasonal averages, which might be better suited to show seasonal differences?
  We chose individual months, because the distribution of the $CO_2$ seasonal signal varies quite fast over time. Many features would already become blurred in seasonal averages (e.g. using DJF, MAM, etc…).

- In the text discussing the figure, you refer to "8-35 km", while the figure uses a pressure scale. Suggest to add a geometrical height scale to the figure, or change the discussion accordingly.
  We have decided to change the text accordingly and now consistently use pressure as altitude coordinate for EMAC model results discussions (see specific comment regarding Line 365).

- It would help adding theta annotations to the contours in every plot
  We added annotations to every panel but only labelled every second level to avoid overloading the plots. also recommend annotating selected wind contours with values, or at least stating the lower threshold in the figure caption. Please also specify in the caption and/or legend whether you considered zonal or horizontal wind speeds.
  We added information about the wind (zonal windspeed) to the legend and the caption (see also next comment).

- Suggest rewording the caption: "...of EMAC CO2 tracer (CO2_MBL_pbl)", "...potential temperature surfaces indicating the UTLS."
  Change as suggested for the former adjustment.
  *"Monthly averaged zonal mean latitude-pressure cross section plots of the diagnostic EMAC $CO_2$ tracer ($CO_2\_MBL\_pbl$) for even months of 2019. The black line is the WMO tropopause. Grey contour lines indicate jet positions (zonal wind speed; lower threshold is 20 m s$^{-1}$, 5 m s$^{-1}$ spacing) and white lines represent selected potential temperature surfaces."* (Caption Fig. 4)

Figure 5.

- Red and orange might be difficult to discern for colourblind readers
  We agree that red and orange might be difficult for colour-blind readers to discern. We tested different versions of the plot with other colour schemes and checked them using the Coblis simulator. However,

we found that this representation is still the best compromise, and we would therefore prefer to keep the figure as it is.

- Caption: "...these pressure levels..."

Done.

- Do the symbols represent individual AirCore flights or averages thereof?

Each symbol represents the results from an individual AirCore flight (average of the observations of that flight for the specified pressure range). To clarify we changed the caption and the text slightly.

*"Symbols represent AirCore measurements from individual flights for comparable pressure ranges."* (Line 418)

*"Temporal evolution (2019-2023) of zonal mean EMAC $CO_2\_MBL\_pbl$ simulation results (lines) at different pressure levels for the 35° N to 55° N latitudinal band with their standard deviation as shaded areas. Symbols represent AirCore observations from individual flights in the vicinity of these pressure levels (mean over the pressure intervals with standard deviation as error bars)."* (Caption Fig. 5)

- Otherwise, trends and seasonal cycle are very well represented here

Figure 6.

- Excellent representation of phase shifts and dampening of seasonal cycle
- Red/green colour contrast might be difficult to read for some users, suggest checking figure with a colourblind simulator or switching to explicitly colourblind-friendly palettes

We switch to a colourblind-friendly palette.

- Can observations be added to this plot, or would that over-clutter it?

It would be very nice to compare this plot directly to a $CO_2$ seasonal signal derived from AirCore measurements and we would have liked to do so. However, for now the seasonal cycle cannot be unambiguously separated from the combined effect of transport and long-term increase in $CO_2$ in observations. This requires further information which is not available. In particular, the lack of $CO_2$-independent transport information for most AirCore flights inhibits the profound isolation of the seasonal $CO_2$ signal from observations.

Figure 8.

- Add subfigure letters a)...f)

Done.

- Consistency: Why are odd months selected here, while even months are shown in Figure 4?

We agree that it would be in principle better to consistently show the same months in Fig. 4 and in Fig. 8 and include the other months in the supplement. We decided to switch between even and odd months because the relevant features that we want to describe in the corresponding paragraphs are better visible in either even months (e.g. $CO_2$ reversed gradient in the SH polar middle stratosphere in Fig 4e) or in odd months (meridional $CO_2\_seas$ gradient differences between end of NH winter vs. NH summer in Fig. 8b and 8d). We would therefore like to keep the selection of the subplots as it is.

- Please give the time range of the climatology in the caption

Done (time range is the same as for Fig. 4: 2000-2023).

*"Zonal mean latitude-pressure cross-sections of the EMAC-derived climatological average (2000-2023) of the $CO_2$ seasonal signal for odd months. The black line indicates the WMO tropopause."* (Caption Fig. 8)

**Technical corrections**

Thank you for reading through the manuscript so carefully and for your detailed feedback. All the spelling inconsistencies (British English vs. American English), typos and grammar have been corrected in the revised version of the manuscript.

"WMO tropopause" spells without a dash; please correct throughout the manuscript (also in the figures).

Done.

Throughout the manuscript, "Figure" needs to be spelled out at the beginning of a sentence.

Done.

Lines 181/182, 202, 204, 232 etc.: Reference instead of inline link

We try to reduce the number of inline links. We simply removed the zendo links for the datasets, since the references are already there. The same applies to the CMIP6 boundary conditions. For the other two cases, we would prefer to keep the inline links, as we think this is not uncommon and we will discuss this with the editorial team during further processing of the manuscript.

*"Moreover, details on the exact methodology and specifications for each individual AirCore flight is given as part of the NOAA and GUF data sets that are available from Baier et al. (2021) and Degen et al. (2025)."* (Line 181)

*"The model setup is based on the EMAC CCMI-2022 setup (Jöckel et al., 2024), but with purely Coupled Model Intercomparison Project Phase 6 (CMIP6) boundary conditions for Ozone Depleting Substances and based on the SSP245 scenario after 2014 (Meinshausen et al., 2017, 2020)."* (Line 203)

Line 211: corresponds

Done.

Line 233: Spelling: "analysed" is British English, "summarized" American English

We decided to use consistently British English.

Line 321: Suggest using "beneath" instead of "below" to clarify that the considered region is situated below 350 hPa in the sense of altitude, not pressure

Done.

Line 324: Again, characteriZed spelled with z, use either AE or BE

Done (BE).

Line 329: no comma after "... vegetation period"

Done.

Line 335: no comma after "... cases"

Done.

Line 340: No return behind "... UTLS well."

We deleted the return before, as the sentence belongs to the previous paragraph.

Line 341: Despite the fact that...

Done.

Line 345: Suggest rewording for clarity: These vertical shifts also appear for other simulated species...

Changed as suggested.

*"These vertical shifts also appear for other simulated species such as $CH_4$ and can be seen in the temperature profiles or the position of the tropopause."* (Line 352)

Line 352: remove comma in "opportunities for analysis, that are"

Done.

Line 381: remove comma after "CO2"

Done.

Lines 414, 415: Suggest writing values with uncertainties like this: (2.44 +/- 0.16) ppm/yr

Done.

Line 435: climatological

Done.

Line 498: simultaneous

Done.

Line 500: remove comma after "both"

Done.

Line 507: The "again" in "definitely more complex again" reads a little awkward here; suggest moving it to the beginning of the sentence: "Again, the short-term variability ..."

Changed as suggested.

*"Again, the short-term variability of the $CO_2$ distribution and of the $CO_2$ seasonal signal is definitely more complex and cannot be resolved by the global EMAC model."* (Line 529)

Line 511: Suggest rewording: "As can be seen from the panels..."

The entire sentence was changed in response to Reviewer #1's reasonable suggestion that the paragraph should not begin by referencing a supplementary figure.

Line 519: "Hadley cell" is spelled without dash

Done.

Lines 517, 522-524 and 541: The parts in brackets disturb the flow; suggest formulating as actual clauses

We rephrased lines 522–524, but we decided to retain the brackets in line 517 with reduced content. The sentence in line 541 was deleted related to a comment from Reviewer #1.

*"The impact of the NH, where the near-surface fluctuation range of the seasonal cycle is 5-10 times higher as in the SH (see peak-to-peak amplitude plot in Fig. S7 for details), extends partly to around 40° S."* (Line 541)

*"In March (end of NH winter) a very pronounced meridional gradient is observed, implicating a distinct transport barrier. In this period, the $CO_2\_seas$ isolines in the STJ region run parallel to the tropopause. In contrast, during July (NH summer) the gradient is much weaker, and the $CO_2\_seas$ isolines intersect the tropopause."* (Line 545)

Line 546: for different latitudes

Done.

**References**

Engeln, A. von and Teixeira, J.: A Planetary Boundary Layer Height Climatology Derived from ECMWF Reanalysis Data, https://doi.org/10.1175/JCLI-D-12-00385.1, 2013.